



# Long-path measurements of pollutants and micrometeorology over Highway 401 in Toronto

Yuan You[1], Ralf M. Staebler[1*], Samar G. Moussa[1], Yushan Su[2], Tony Munoz[2], Craig Stroud[3], Junhua Zhang[3], and Michael D. Moran[3]

[1]Air Quality Processes Research Section, Environment and Climate Change Canada, Toronto, Ontario, Canada, M3H 5T4.
[2]Ontario Ministry of the Environment and Climate Change, Toronto, Ontario, Canada, M9P 3V6
[3]Air Quality Modelling and Integration Section, Environment and Climate Change Canada, Toronto, Ontario, Canada, M3H 5T4.

*Correspondence to: ralf.staebler@canada.ca

## Abstract

Traffic emissions contribute significantly to urban air pollution. Measurements were conducted over Highway 401 in Toronto, Canada, with a long-path Fourier Transform Infra-Red Spectrometer (FTIR) combined with a suite of micrometeorological instruments, to identify and quantify a range of air pollutants. Results were compared with simultaneous in-situ observations at a roadside monitoring station, and with output from a special version of the operational Canadian air quality forecast model (GEM-MACH). Elevated mixing ratios of ammonia (0-23 ppb) were observed, of which 76 % were associated with traffic emissions. Hydrogen cyanide was identified at mixing ratios between 0 and 4 ppb. Using a simple dispersion model, an integrated emission factor of on average 2.6 g km$^{-1}$ carbon monoxide was calculated for this defined section of Highway 401, which agreed well with estimates based on vehicular emission factors and observed traffic volumes. Based on the same dispersion calculations, vehicular average emission factors of 0.04, 0.36 and 0.15 g km$^{-1}$ were calculated for ammonia, nitrogen oxide, and methanol respectively.

## 1.     Introduction

In 1996, 45.2 % of the population of Toronto, Canada's largest city, lived within 500 m of a highway or within 100 m of a major road (HEI, 2010). This percentage was updated to 40 % in 2002 and 2005 (Su et al., 2015). Therefore, a significant portion of the population is exposed to traffic-related air pollution. Pollutants that have been previously reported from motor vehicles include nitrogen oxides ($NO_x$), carbon monoxide (CO), ultrafine particles and $PM_{2.5}$, black carbon, and volatile organic compounds (VOCs) (Brugge et al., 2007; Zhou and Levy, 2007; Karner et al., 2010). Motor-vehicle-related emissions contributed about 40 % of the $PM_{2.5}$ in Toronto during 2000 to 2001 according to Lee et al. (2003). A study on a global scale indicated that traffic emissions are important contributors to outdoor air pollution (ozone ($O_3$) and $PM_{2.5}$) associated with premature mortality in 2010 for the U.S.A., Germany, and the U.K. (Lelieveld et al., 2015).



Exposure to these air pollutants is associated with negative health effects. Laboratory studies have indicated that inhalation

of fine particles and $O_3$ even for a short time causes acute conduit artery vasoconstriction (Brook, 2002). Studies in Toronto have shown that exposure to traffic-related air pollution is associated with respiratory conditions (Buckeridge et al., 2002), increased risk of circulatory mortality (Jerrett et al., 2009), cardiovascular mortality (Chen et al., 2013), ischemic heart disease (Beckerman et al., 2012), and childhood atopic asthma (Shankardass et al., 2015). Research results in other locations have also shown associated negative health effects, such as asthma (Lin et al., 2002; McConnell et al., 2006), cancer and

leukemia (Pearson et al., 2000) of children, and development of obesity in children (Jerrett et al., 2014). Exposure to traffic-related air pollution may also be associated with increased risk of dementia. Chen et al. (2017) studied a large adult population of Ontario between 2001 and 2012, and they found that incident dementia was 7 % higher for people living within 50 m away from major roads than for the general population.

The segment of Highway 401 crossing Toronto is the busiest highway in North America, with annual average daily traffic

(AADT) counts of 410,000 (Ontario Ministry of Transportation, 2013). A few studies on air pollution have been conducted near Highway 401 in the Greater Toronto Area in the past. Beckerman et al. (2008) measured air pollutants at the same location as the current study presented here. They showed elevated nitrogen dioxide ($NO_2$) and VOCs levels both upwind and downwind of Highway 401, and pollutants did not decay to background levels until 300-500 m downwind.

The main focus of this study was to measure gaseous pollutants through the use of a long-path Fourier Transform Infra-Red

(FTIR) spectrometer. Compared to off-line post analytical methods, FTIR can measure mixing ratios of a variety of gaseous pollutants in near-real time simultaneously, without a container or tubing and without experimental contamination after sampling. Another advantage of FTIR is that it retrieves path-averaged mixing ratios instead of point measurements, so it is less dependent on wind direction. A common approach to retrieve mixing ratios of species from FTIR measurements is to compare the measured spectra with reference spectra obtained in the laboratory at a given temperature and pressure with a

known mixing ratio. The Pacific Northwest National Laboratory (PNNL) established a database of gas-phase infrared spectra for pure compounds (Sharpe et al., 2004). Another source of reference spectra is the molecular absorption database HITRAN (HIgh resolution TRANsmission molecular absorption database) (Rothman et al., 1998, 2013). A major weakness of FTIR is the interference from water vapour, which can be too strong for some species and some absorption features, for example when quantifying mixing ratios of nitrogen oxide (NO) and $NO_2$ in a humid environment.

FTIR spectrometry has been used to quantify the mixing ratios of various trace species emitted by forest biomass burning (Yokelson et al., 1996, 1997; Yokelson, 1999; Akagi et al., 2014; Paton-Walsh et al., 2014; Smith et al., 2014), volcanoes (Horrocks et al., 1999), industrial parks (Wu et al., 1995), and urban areas (Hong et al., 2004). Vehicle emissions have also been investigated by long-path FTIR. Bradley et al. (2000) performed a three-hour measurement in the morning beside a road in Denver using long-path FTIR and quantified mixing ratios of CO, carbon dioxide ($CO_2$), and nitrous oxide ($N_2O$).





There are very few studies, however, that combine direct measurements of mixing ratios of gas-phase pollutants from highway emissions with detailed information on the micrometeorology at the same time and same location. Micrometeorological conditions will be shown here to have a significant effect on modulating the observed mixing ratios. Baldauf et al. (2008) studied the effect of traffic emission and meteorological conditions on the local air quality near a road in Raleigh, North Carolina, U.S.A. in 2006 using long-path FTIR. The CO and $NH_3$ mixing ratios reached their peaks around

7:00, corresponding to the morning commuter rush hour. Another peak of CO around 17:00 corresponded to a traffic peak as well. Their results showed that horizontal turbulence intensities were large between noon and 17:00 when measured pollutant mixing ratios were low. Brachtl et al. (2009) measured polycyclic aromatic hydrocarbon mixing ratios along with CO, $SO_2$, $NO_x$, and $PM_{2.5}$ at the side of a road for four days in Quito, Ecuador. An early morning peak followed by a sharp drop of mixing ratios was observed, corresponding to the sharp increase of solar irradiation after 7:00. Their results also

showed another weaker peak of CO, $NO_x$ and $PM_{2.5}$ between 20:00 and 21:00, after solar irradiation decreased to zero and temperature dropped. Other studies monitored ambient temperature and wind speed to understand meteorological and mixing conditions, and changes in pollutant mixing ratios were found to correlate with these conditions. Gentner et al. (2009) measured CO and VOC mixing ratios 1 km from a highway for two month-long periods in Riverside, California in 2005. They attributed the minimum CO mixing ratios observed in the afternoon to increased mixing and dilution. Durant et

al. (2010) presented one-day measurements of pollutant mixing ratios, wind speed, and ambient temperature, along with traffic density. They observed an increase of pollutant levels before sunrise and a sharp decrease after sunrise. Hu et al. (2009) monitored pollutant mixing ratios, wind, and ambient temperature in the early morning period for three days. They found mixing ratios were much higher before sunrise even though traffic volume was lower than later during the daytime.

    In this study we conducted measurements of gaseous pollutants, along with detailed micrometeorological conditions,

continuously over two weeks from July 16 to July 31, 2015 across Highway 401 in Toronto. Quantified pollutants which were discussed in the text included: CO, $NH_3$, $O_3$, formaldehyde (HCHO), hydrogen cyanide (HCN), and methanol ($CH_3OH$). In addition, we used the proximity of a NAPS (National Air Pollution Surveillance) surface measurement station, which was located near the middle of the FTIR path next to the highway, to conduct an in-depth comparison of a range of pollutants measured by both the path-integrating FTIR instrument and the in-situ station. To our knowledge, this is the first

direct comparison of this kind to be published. These data are then used to evaluate a research version of the GEM-MACH (Global Environmental Multiscale model-Modelling Air quality and CHemistry) air quality forecast model (Moran et al., 2010; Gong et al., 2015; Makar et al., 2015a). Finally, highway-integrated emission rates are estimated using a backward Lagrangian stochastic dispersion model, and compared with previously published engine emission results scaled by traffic volume.

The objectives to be addressed with this analysis are: 1) to evaluate the capabilities of the long-path FTIR spectrometer for quantifying the mixing ratios of gaseous pollutants; 2) to quantify gaseous-pollutant mixing ratios as a function of traffic



volume and micrometeorological conditions; 3) to compare mixing ratios from these direct measurements to GEM-MACH model results; and 4) to evaluate the feasibility of deriving emission rate estimates from these measurements using an inverse dispersion model.

## 2.   Experimental

### 2.1   Long-path FTIR setup and analysis

As shown in Fig. 1, the FTIR and scintillometer instruments were set up on the south side of Highway 401 at 125 Resources Road (43.711$^{o}$N, 79.543$^{o}$W) in Toronto, Ontario, Canada.  Our study used a commercial Open Path FTIR spectrometer (OPS, Bruker, Germany).  The infrared source is an air-cooled Globar at a temperature between 1200 and 1500$^{o}$C.  The
emitted radiation is directed with a lens through the interferometer where it is modulated, travels along the measurement path across the highway, reaches a retroreflector array that reflects the radiation back, travels back across the highway, and enters a Stirling-cooled mercury cadmium telluride (MCT) detector. The FTIR spectrometer was set up on the roof of a building, about 8 m above the ground, while the retroreflector array was mounted on a mast at 4 m above ground level north of Highway 401. The distance between the spectrometer and retroreflector array was 310 m, resulting in a path length of 623.7
m, which includes 3.7 m of internal reflections. The fraction of the path that was directly over the highway was 117 m (Fig. 1).

In this study, spectra were measured at a resolution of 0.5 cm$^{-1}$ with 250 scans co-added to increase signal-to-noise ratio, resulting in roughly a one-minute temporal resolution.  Before July 23$^{rd}$, 100 scan co-adds were used.  At the beginning of the measurement period, a straylight spectrum was recorded by pointing the spectrometer away from the retroreflector.  This
straylight spectrum accounts for radiation entering the spectrometer from sources other than the retroreflector array and was subtracted from all the measurement spectra before performing further analysis. The Bruker OPS software analyzes spectra to derive the mixing ratios of the gases of interest using a quantitative method described in detail in Griffith (1996).  Spectral ranges for retrieval analysis for each target gas were chosen based on prominent absorption features of the target gas and spectral windows as found in previous studies.  Reference spectra were fitted to the measured spectra using classic least
squares (CLS) methods within the chosen window.

For each gas of interest, a reference file was made including spectra of target and interference gases.  High-resolution reference spectra at 296 K and 1013.25 hPa were taken from the HITRAN database when available.  For species not available in the HITRAN database, the reference spectra were taken from the PNNL database.  Spectral ranges for fitting, interference gases, and detection limit based on Bruker's results for each pollutant retrieved in this work are listed in Table 1.
Raw estimates of mixing ratios of gases of interest were retrieved assuming an ambient temperature of 296 K and air



pressure of 1013.25 hPa. These values were then corrected for the actual temperature and pressure measured at the NAPS station using the ideal gas law.

The air temperature also has a secondary effect on the signature of the IR spectrum of individual gases. The population of the higher vibrational and rotational states increases as temperature increases. However, the sensitivity of temperature on

those signatures depends on the individual gas and the range of temperature change. Reference spectra at different temperatures are available for a limited number of species at 278, 298, and 323 K. Temperature-dependent reference files can be made in the software to combine reference spectra at these three temperatures. To test the effect of temperature on the retrieved mixing ratio, spectra during the last eight days of July were analyzed for $NH_3$, $CH_4$, CO, and $CO_2$ by using these temperature-dependent reference files. The maximum difference in retrieved mixing ratio for the $45\,^{o}C$ range is 8.9 % for

$NH_3$, 4.2 % for $CH_4$, 8.3 % for CO, and 4.1 % for $CO_2$. Based on this test, we estimate that using reference spectra at standard temperature and pressure contributed to an uncertainty of less than 10 % in the final mixing ratio results.

## 2.2    Scintillometer theory and setup

Simultaneous long-path turbulence measurements were made using a Boundary Layer Scintillometer (BLS 900, Scintec, Germany). The scintillometer receiver was set up next to the FTIR spectrometer, on the south side of Highway 401 (Fig. 1).

The transmitter, with two disks of 924 LEDs emitting at 880 nm, was set up on the north side of Highway 401 just above the FTIR retroreflector. The mean height of the scintillometer path was 8 m above ground level. In this study, the LEDs were operated in continuous mode. Radiation is directed onto the photodiodes in the receiver, which quantify the turbulence-induced fluctuations in the optical refractive index between the transmitter and receiver.

The structure function of the optical refractive index $C_n^2$ is determined from the fluctuations of the light-beam intensities

received by the scintillometer receiver. The structure parameter of temperature $C_T^2$ can be derived from $C_n^2$ given the ambient temperature, humidity, pressure, and wavelength (Thiermann and Grassl, 1992). The determination of the sensible heat flux H based on $C_T^2$ needs additional assumptions based on Monin-Obukov Similarity Theory (MOST).

H is defined as

$$H = -\rho C_p \theta_* u_* \tag{1}$$




where ρ is the air density, $C_p$ is the specific heat capacity of air at constant pressure, $\theta_*$ is the temperature scale, and $u_*$ is the friction velocity. $\theta_*$ and $u_*$ are determined by the MOST functions (Wood et al., 2013):

$$\frac{C_T^2 z^{2/3}}{\theta_*^2} = \Psi_H\left(\frac{z}{L}\right) \tag{2}$$

$$\frac{\kappa z U(z)}{\ln\left(\frac{z}{z_0}\right) u_*} = \Psi_M\left(\frac{z}{L}\right) \tag{3}$$

where the Obukov length L is defined as

$$L = \frac{T}{\kappa g}\frac{u_*^2}{\theta_*} \tag{4}$$

and z is the height above the surface, $z_0$ is the roughness length, the von Kármán constant $\kappa=0.4$, g is the gravitational constant, and U(z) is the mean horizontal wind speed. The scaling functions $\Psi_H$ and $\Psi_M$ can be calculated by Eqs. (5) and (6) (Thiermann and Grassl, 1992) and Eqs. (7) (Paulson, 1970) and (8) (Businger et al., 1971):

$$\Psi_H = 4\beta_1\left[1 - 7\frac{z}{L} + 75\left(\frac{z}{L}\right)^2\right]^{-1/3} \qquad for\ \frac{z}{L} < 0 \qquad (5)$$

$$\Psi_H = 4\beta_1\left[1 - 7\frac{z}{L} + 20\left(\frac{z}{L}\right)^2\right]^{1/3} \qquad for\ \frac{z}{L} > 0 \qquad (6)$$

with  $\beta_1=0.86$

$$\Psi_M = -2\ln\left(\frac{1+\chi}{2}\right) - \ln\left(\frac{1+\chi^2}{2}\right) + 2\arctan(\chi) - \frac{\pi}{2} \qquad for\ \frac{z}{L} < 0 \qquad (7)$$

where  $\chi = \left(1 - 15\frac{z}{L}\right)^{1/4}$

and  $\Psi_M = 4.7\frac{z}{L} \qquad for\ \frac{z}{L} > 0 \qquad (8)$

These calculations were performed with the software provided by Scintec (SRun, version 1.31; see http://www.scintec.com/english/web/scintec/Details/A012000.aspx). The procedure to calculate the sensible heat flux H from measurements is as follows:




An initial |L|=|L$_{ini}$| = 1000 m is assumed, where the signs of L and H are determined by the simultaneous measurement of the vertical temperature gradient ΔT/Δz. Then $\Psi_H$ is calculated from Eqs. (5) and (6) using L and path height z. $\theta_*$ can then be calculated with $C_T^2$ using Eq. (2). Next $\Psi_M$ is calculated from Eqs. (7) and (8). Friction velocity $u_*$ is then calculated by Eq. (3) given the measured wind speed (from the NAPS station) and z$_0$. A new L can then be calculated from Eq. (4) using the calculated $\theta_*$ and $u_*$. This process is then repeated iteratively until the change in L is smaller than 0.1. Finally, H is calculated from Eq. (1) using the last calculated values for $\theta_*$ and $u_*$.

z/L is a surface-layer scaling parameter describing the dynamic stability of the surface layer (Stull, 2003). Negative z/L values indicate an unstable surface layer, while positive z/L values indicate a stable surface layer. The closer the value of z/L is to zero, the closer conditions are to neutral stability. In this work, H and z/L were used to determine the strength of turbulence and mixing in the surface layer. Solar radiation data were taken from a York University weather station (http://www.yorku.ca/pat/weatherStation/index.asp) situated about 9 km north-west of our site. We used the downwelling

short wavelength radiation data to quantify cloudiness during the study. During the 16-day measurement period, only July 17[th] had some rain, and all other days were mostly sunny.

### 2.3 NAPS measurements

The NAPS program aims to provide accurate and long-term air quality data with uniform standards across Canada by coordinating the data collection from existing air quality monitoring networks (Galarneau et al., 2016). The first NAPS

measurements were in 1972, focusing on $SO_2$ and particulate matter. Currently, $SO_2$, CO, $NO_2$, $O_3$, and $PM_{2.5}$ are continuously measured at more than 200 sites across Canada (Environment and Climate Change Canada, http://www.ec.gc.ca/rnspa-naps/default.asp?lang=En&n=8BA86647-1, last accessed March 25, 2017). The NAPS data shown in this study come from the trailer located right beside the FTIR path on the south edge of the Highway 401 (see Fig. 1). Measurements include CO, $CO_2$, NO, $NO_2$, $O_3$, $SO_2$, and $PM_{2.5}$ at one-minute temporal resolution, as well as

meteorological parameters including air temperature, pressure, relative humidity, and wind speed and direction.

### 2.4 GEM-MACH model

GEM-MACH is a chemical transport model embedded within the GEM (Global Environmental Multiscale) numerical weather forecast model of Environment and Climate Change Canada (ECCC) (Côté et al., 1998a,b). Meteorological conditions (Makar et al., 2015b) and air quality processes, including gas-phase, aqueous-phase, and heterogeneous chemistry

and size-resolved aerosol processes, are included in GEM-MACH (Moran et al., 2010; Gong et al., 2015; Makar et al., 2015a). GEM-MACH is used operationally by ECCC for short-term air quality forecasting on a North American grid with 10-km horizontal grid spacing (Moran et al., 2014; Pavlovic et al., 2016). In this study, a research version of GEM-MACH





simulated concentrations of pollutants with a horizontal grid-cell size of 2.5 km within a 40 m layer above ground level. Our measurement site was located within one model grid cell. Hourly outputs were obtained from GEM-MACH in this study.

GEM-MACH outputs instantaneous pollutant mixing ratio fields once an hour, including CO, $O_3$, $NH_3$, HCHO, NO, and $NO_2$. The FTIR and the NAPS measured mixing ratios once a minute. In order to compare model results and measurements for similar periods, GEM-MACH results were averaged over the two bracketing timestamps to get an estimate of the average mixing ratio of pollutants over each hour, while the measurements results were averaged every hour to match the temporal resolution of GEM-MACH results.

**2.5     WindTrax estimation of source emission rate from mixing ratio measurements**

Various approaches have been developed to deduce source emission rates from pollutant concentrations, including inverse dispersion models (cf. Flesch et al., 2004). The WindTrax model (http://www.thunderbeachscientific.com/) calculates the emission rate $Q$ by the formula

$$Q = \frac{(C - C_b)}{(C/Q)_{sim}} \qquad (9)$$

where C is the concentration of a pollutant at the measurement site, $C_b$ is the background concentration, and $(C/Q)_{sim}$ is the simulated ratio of concentration at the site to the emission rate. In this study, $(C/Q)_{sim}$ is calculated using a backward Lagrangian Stochastic (bLS) model (Flesch et al., 1995). In the bLS model, numbers of virtual particles are released at the site, and individual upwind trajectories are calculated backward in time from the site. Then the fraction of trajectories that originate from the source area is determined. WindTrax can handle complex source-area shapes but not variations in

topography. The micrometeorological condition inputs for the bLS model are $u_*$ and L obtained from the scintillometer measurements as well as wind direction and ambient temperature from the NAPS trailer.

In this study, we used the mixing ratio of CO measured by the FTIR to estimate the CO emission rate from the highway (a "bottom-up" approach). The estimated emission rates are then compared to the emission rates derived from traffic volume combined with emission factors of vehicle engines (Section 3.6). These results will help evaluate the capability of deducing

emission rates from our measurements.

**2.6     Traffic volume data**

Traffic volume data were provided by the Ontario Ministry of Transport, in units of vehicles per hour passing a point on Highway 401. Before July 20[th], counts were available at the Islington Avenue intersection, about 700 metres to the west of our site. However, after July 20[th], data at Islington Avenue were not available, and we instead used traffic volume data at a



nearby intersection to the east of our site at Avenue Road, which showed a linear relationship with traffic volumes at

Islington Avenue. Therefore, the traffic volume data before July 20[th] were measured at the Islington Avenue intersection,

while data after July 20[th] for the same location are estimated.

## 3.    Results and discussion

### 3.1    Micrometeorology

During the study, the mean wind speed measured at the NAPS trailer was 2.5 m s$^{-1}$, with a range from 0 to 9.9 m s$^{-1}$ and

quartiles of 1.3 and 3.3 m s$^{-1}$. The mean friction velocity $u_*$ during the study was 0.40 m s$^{-1}$, with a range from 0.02 to 1.31 m

s$^{-1}$ and quartiles of 0.25 and 0.52 m s$^{-1}$. The mean ambient temperature was 24$^{o}$C, with a range from 14 to 33$^{o}$C.

In Toronto in late July, sunrise occurs at about 6:00 EDT (Eastern Daylight saving Time, same time labels were used for the

entire study), solar noon occurs at about 13:30, and sunset occurs at about 21:00.  As shown in Fig. 2, sensible heat flux H

started to increase beginning at 6:00, reached its maximum in the early afternoon around 13:30, and then decreased to its

minimum after 23:00.  The downwelling shortwave solar radiation started to increase at 6:30 and reached a peak around

13:00.  It is notable that H remained positive throughout the night and started to increase before sunrise.  We surmise that

this is due to traffic providing a source of sensible heat and mechanical and convective turbulence, as well as slow release of

heat from the pavement at night (Sailor and Lu, 2004; Khalifa et al., 2016).  A rough estimation of  33 W m$^{-2}$ (56 % of H) at

5:30 (before the sunrise) was contributed by vehicles on the highway, based on the traffic volume, the ratio of energy loss

from gasoline engine, and the typical fuel consumption of gasoline vehicles. Result of z/L remained negative throughout the

night, indicating that the surface layer was always unstable or neutral. $u_*$ also varied diurnally, with higher values from 08:00

to 21:00 and lower values during the night, which also suggests stronger turbulence in the daytime and is correlated with H.

All of these measurements show that mixing and turbulence started to increase quickly after sunrise, reached a maximum in

the early afternoon, and decreased to a minimum after 23:00.

### 3.2    CO

#### 3.2.1    Comparison between FTIR, NAPS and GEM-MACH

CO is directly emitted by vehicles, and CO emission factors from vehicles have been reported in various studies. Among

those studies, Bradley et al. (2000) and Baldauf et al. (2008) measured CO from traffic by open-path FTIR. CO has also been

used as a reference pollutant to determine emission factors of other primary pollutants by calculating concentration ratios of

pollutants to CO (Warneke et al., 2007; Baker et al., 2008; Gentner et al., 2013). As shown in Fig. 3, mixing ratios of CO

from the FTIR and the NAPS generally agree with each other, but with a significant offset and amplitude difference. The



GEM-MACH simulation predictions and the measurements of CO mixing ratio agree well in general (Fig. 3). The GEM-MACH simulated most of the peak mixing ratios consistent with measurements.

One contributing reason for the differences of CO mixing ratios between the FTIR and the NAPS may be that the FTIR and the NAPS were not sampling the exact same air, i.e., the measurements represented different footprints. The FTIR measured the air along the path across and above the Highway 401, which always included some pollutants emitted from traffic. In contrast, NAPS numbers represented point measurements beside the south edge of the highway. Since CO is directly emitted from the highway, CO mixing ratios from the NAPS might be expected to be lower than those from the FTIR,

particularly when the wind is from the south and towards the highway. The polar plot in Fig. 4a clearly shows the dependence of the CO mixing-ratio difference between the FTIR and the NAPS on wind direction. When the wind came from the north over the highway towards the NAPS trailer (above the dashed line), CO mixing ratios from the FTIR were still greater than CO from the NAPS most of the time, but the differences were much smaller than for other wind directions.

Spatial incommensurability remains an issue when comparing gridded air quality model predictions with measurements. A

GEM-MACH surface-level mixing ratio represents a mean value over a grid-cell volume that is 2.5 km by 2.5 km by 40 m in size whereas the FTIR measurements are averages over a line that is an order of magnitude shorter than the length of the side of GEM-MACH grid cell and the NAPS measurements correspond to values at a single point right at the south edge of the highway. In addition, the emissions considered by GEM-MACH for a particular grid cell include the contributions of all point, line, area, and volume sources contained within that grid cell, and the sum of these multiple sources is assumed to be

distributed uniformly across the grid cell (see Fig. S1). Thus, the artificial mixing and dilution of emissions within a model grid cell, subgrid-scale variations in wind direction, and the locations of emissions sources relative to measurement locations may impact the comparison between model results and measurements, particularly for primary pollutants.

To investigate the effect of wind direction on the difference of CO mixing ratios between from NAPS and GEM-MACH, the difference was plotted as a function of wind direction (Fig. 4b). GEM-MACH predictions were lower than the NAPS

measurements when the wind direction was from the highway towards the NAPS trailer (cold colors), while GEM-MACH predictions were greater than the NAPS measurements when the wind was from other directions (warm colors). A linear regression analysis of CO mixing ratios from GEM-MACH and NAPS stratified by wind direction is shown in Fig. S2. The slope of the best-fit line when the wind was from the highway to the trailer was less than 1.0 and the mean bias was negative; for winds from other directions, the slope was greater than 1.0 and the mean bias was positive. These results are consistent

with the above discussion about point-measurement representativeness vs. model grid-cell averages. When the wind was from the highway, CO measurements by the NAPS were directly influenced by the trailer's close proximity to heavy traffic emissions, a subgrid-scale emissions feature that could not be well represented by the air quality model.



### 3.2.2 Average diurnal cycles

During weekdays (Fig. 5), the minimum traffic volume was about 5000 vehicles h$^{-1}$ between 2:00 and 5:00; traffic started to

increase after 5:00 and reached a maximum 23800 vehicles h$^{-1}$ from 7:00 to 8:00. After reaching the morning peak, traffic

volume remained high through most of the day, starting to drop after 21:00. The CO mixing ratio on weekdays rapidly

reached a peak between 6:00 and 8:00. H and u$_*$ during this period were still low compared to the middle of the day (see

Fig. 2), indicating that turbulence was weak compared to the afternoon. This suggests that the peaks of CO mixing ratio

observe in the early morning were due to rapid increase and accumulation of emissions of CO while there was still little

convection, before stronger mixing started later in the morning. Similar observations have been previously reported (Janhäll

et al., 2006; Hu et al., 2009; Durant et al., 2010). In the afternoon on weekdays, when the traffic volume was still high, the

CO mixing ratio dropped significantly, compared to early morning rush hour. Turbulence was strong at noon and in the

afternoon, so emitted pollutants were diluted efficiently. Therefore CO mixing ratio in the afternoon was lower than in the

morning despite similar traffic volumes. In the late evening (21:00 to 0:00), there was a secondary peak in mixing ratios of

primary pollutants from traffic even as traffic volume started to drop, again due to diminished vertical mixing leading to

accumulation in the surface layer after sunset (Gentner et al., 2009).

The average weekday diurnal cycles of CO, traffic volume and turbulence/mixing clearly show that turbulence and mixing

played an important role on the mixing ratios of primary pollutants above the highway. On weekends, traffic volume

increased more gradually during the morning until plateauing around 11:00 and on average remained high until after 22:00.

The diurnal patterns of CO mixing ratio were flatter but with greater variability, compared to weekdays. The median CO

mixing ratio on weekends was close to that on weekdays, except for the early morning period. These comparable CO levels

for weekdays and weekends for comparable traffic volumes suggest that traffic was the main emission source of CO.

Ambient temperature may also affect emissions from vehicles and hence pollutant mixing ratios near traffic (Choi et al.,

2010; Rubin et al., 2006). However, since the range of ambient temperatures was small during the study period (from 15 to

32$^{\circ}$C), the effect of temperature on the average diurnal cycle of CO mixing ratio was likely weak.

### 3.3 NH$_3$

NH$_3$ can form secondary aerosols that are associated with negative health effects (Seinfeld and Pandis, 2006; Behera and

Sharma, 2012; Liu et al., 2015) as well as radiative forcing impacts. According to the U.S. Environmental Protection

Agency (U.S. EPA)'s trends data for 2016, 2.4 % of U.S. national NH$_3$ emissions are from vehicles which are more

important sources in urban regions. After the three-way catalytic converter (TWC) was introduced to gasoline vehicles in

1981 and became used widely, NH$_3$ (as a product formed in TWC from the reaction of NO with CO and H$_2$O) emissions

from vehicles increased (Moeckli et al., 1996; Fraser and Cass, 1998; Kean et al., 2000). NH$_3$ is also involved as a reagent in





the reduction processes for NO in selective catalytic reduction converters (SCR) in diesel vehicles. Therefore, diesel vehicles could also contribute to $NH_3$ emissions, due to the aging of catalysts and over-doping of urea. However, they play only a minor role in $NH_3$ traffic emissions compared to gasoline vehicles based on the emission inventory used by GEM-MACH over Greater Toronto and Hamilton Area (ECCC, 2014). $NH_3$ is gaining importance as a pollutant from traffic due to the gaining use of emission control systems, but previous studies which directly measured $NH_3$ mixing ratio from traffic are rare. Elevated mixing ratios of $NH_3$ between 0 and 23 ppb were observed with the FTIR in this study (Fig. 6). Baldauf et al. (2008) showed diurnal plots for traffic volume and mixing ratios of $NH_3$ measured by open-path FTIR 20 m and 300 m from a main road. The $NH_3$ mixing ratio they reported was to be between 10 and 35 ppb, comparable to our results.

Traffic emissions appear to be very important to $NH_3$ in urban environments, although residential garbage collection (Reche et al., 2012), soil and fertilizers, biomass burning, natural ecosystems, sewage and landfill, and direct emissions by humans and animals could also contribute (Sutton et al., 2000). Yao et al. (2013) found a good linear correlation between mixing ratios of $NH_3$ and NO during periods in the morning at the same site beside Highway 401. CO has been used as a common reference pollutant from vehicle emissions as discussed in Section 3.2.1, and studies have also used [$NH_3$] / [CO] ratio to correlate $NH_3$ to traffic emissions. A linear correlation between emission factors of CO and $NH_3$ was found in light and medium-duty vehicles in the California South Coast air basin by Livingston et al. (2009). Perrino et al. (2002) found a linear relationship between mixing ratios of $NH_3$ and CO at a traffic site in Rome. A linear relationship between $NH_3$ and CO mixing ratios from the FTIR over the whole period was also observed in this study (slope=0.023, $r^2$=0.60). This linear relationship suggests that $NH_3$ and CO shared a common source, which in this case, a significant fraction (76% during morning rush hour: see discussion in the next paragraph) of $NH_3$ came from traffic. The slope of 0.023 ([$NH_3$] / [CO]) is close to values previously reported (Livingston et al., 2009). $NH_3$ emission factors from vehicles in the literature are in the range of 0 to 0.144 g km$^{-1}$ depending on various factors such as fuel type, driving cycle, vehicle engine power, engine temperature, and catalyst aging (Durbin et al., 2002; Huai et al., 2003). Therefore, differences in slopes among studies are to be expected.

Average diurnal cycles of $NH_3$ mixing ratios on weekdays and weekends are shown in Fig. 6. To estimate the $NH_3$ due to traffic emissions, it was assumed that all $NH_3$ emissions from traffics are correlated with CO. Thus, a background CO mixing ratio of 265 ppb was subtracted from the retrieved CO mixing ratio and the result was regressed against $NH_3$ mixing ratios, resulting in a traffic-related $NH_3$ being estimated as 0.023×([CO]_FTIR- 265) ppb. The 265 ppb CO background was the intercept of CO from the linear regression of $NH_3$ with CO. The resulting weekday and weekend diurnal cycles of the estimated $NH_3$ mixing ratio due to traffic emissions are plotted in Fig. 5. During the morning rush hour and late at night on weekdays, traffic emissions contributed more to $NH_3$ levels than during other times of day. On weekends, the diurnal variation of total $NH_3$ was weaker, and estimated $NH_3$ from traffic accounted for essentially all $NH_3$ observed. Overall, there is no indication of a background offset of $NH_3$, and most measured $NH_3$ at this site can be accounted for by traffic emissions.



NH$_3$ measurements by the FTIR also agreed well with GEM-MACH model simulations (Fig. 6). The analysis results of the

traffic contribution to NH$_3$ around the site based on the FTIR measurements are consistent with the GEM-MACH model

NH$_3$ input emissions, which show that the main source of NH$_3$ at this location is vehicular (Fig. S1).

### 3.4  O$_3$, NO, NO$_2$, and HCHO

O$_3$ is a secondary pollutant and is not emitted directly by vehicles. NO reacts with O$_3$ forming NO$_2$ on a time scale of a few

minutes during the day. Photochemistry between VOCs and ambient oxidants produces O$_3$, and HCHO is one of the products

from these photochemical reactions. The chemistry of titration and photochemical production of O$_3$ has been discussed

previously in detail (Marr and Harley, 2002b; Fujita et al., 2003; Seinfeld and Pandis, 2006; Murphy et al., 2007). The time

series of O$_3$ mixing ratio from the FTIR also agrees broadly with the NAPS O$_3$ measurements (Fig. 3).  However, the polar

plot in Fig. 7a shows that O$_3$ mixing ratios measured by the FTIR and the NAPS were close when the wind was from the

highway, whereas O$_3$ from the FTIR was much lower than O$_3$ from the NAPS, when the wind was from other directions.

These results can be explained by the titration of O$_3$ over the highway by NO emissions from vehicles: when the wind is

from the north, the O$_3$ reaching the NAPS trailer has been titrated, but when the wind is from the south, O$_3$ measured at the

NAPS site is titrated over the highway downwind of the measurement point.

Mixing ratios of NO and NO$_2$ can be retrieved from the FTIR spectra, but the correlation coefficients of fitting are less than

0.1 and estimated mixing ratios contain large offsets and biases, probably due to the strong interference from water vapor.

Therefore, mixing ratios of NO and NO$_2$ from the FTIR are not shown here.  The GEM-MACH simulations and NAPS

measurements for NO and NO$_2$ often do not agree well (Fig. 8). The disagreements can again be partially explained by the

influence of wind direction.  Like CO and NH$_3$, NO is directly emitted from vehicles, but it reacts in the atmosphere much

more quickly than CO or NH$_3$.  Polar plots for NO and NO$_2$ (Figs. 7b and 7c) show the effect of wind direction on the

mixing-ratio differences between GEM-MACH results and NAPS measurements.  When the wind blew from the NAPS

trailer towards the highway, the difference was close to zero, but when the wind blew across the highway towards the trailer,

GEM-MACH predictions were significantly lower than NAPS measurements. Similar to the CO comparison, the NAPS

measurements were strongly influenced by traffic emissions when the wind came from highway compared to GEM-MACH.

Note that GEM-MACH simulates mean pollutant mixing ratios within a 40-m layer whereas the FTIR path was located

about 8 m above the highway pavement and the inlet of the NAPS trailer was about 3 m above the ground.  These different

heights also contribute to the disagreement between measurements and model results.

The NO$_x$ (nitrogen oxides, NO$_x$ = NO+NO$_2$) mixing ratio measured at the NAPS station on weekdays showed a similar

average diurnal cycle (Fig. 9) to CO by the FTIR, reaching a peak over 100 ppb from 6:00 to 8:00 followed by significant

decrease in the middle of the  and a secondary peak between 20:00 and 23:00.  The diurnal cycle of NO$_x$ on weekends with





mixing ratios of 0-35 ppb over the whole day, significantly lower than weekday $NO_x$ levels, was also less variable. Reduced $NO_x$ levels on weekends may have been due to fewer diesel vehicles operating on weekends; this pattern has been reported in studies in California (Marr and Harley, 2002a; Harley et al., 2005; Kim et al., 2016). Zhang et al. (2012) found that fewer diesel vehicles were observed on weekends on another major highway in the Toronto area. The annual sales of fuel used for on-road motor vehicles in Canada in 2015 were 42.6 billion litres of gasoline and 18.0 billion litres of diesel (Statistics

Canada 2016), i.e., a significant fraction of fuel burned is diesel. Therefore, lower diesel vehicle volumes on weekends may have contributed to different emissions of $NO_x$ on Highway 401 near our site. The $[NO_x] / [CO]$ ratio also has been used to check the chemical conditions related to $O_3$ production. Figure 10 shows that the ratios in this study are 0.20 and 0.10 for weekdays and weekends, respectively. The lower ratio on weekends is likely due to reduced numbers of diesel vehicles, which is consistent with a previous study by Kim et al. (2016). Our $[NO_x] / [CO]$ ratios during both weekdays and weekends

are greater than their results (0.11 and 0.033), but our study only focused on near-surface observations over a short defined section of Highway 401 while their observations were at 1 km above the ground level with a bigger footprint which included off-road emissions and other local sources.

Figure 9 also shows average weekday and weekend diurnal cycles for $O_3$ measured at the NAPS station. One interesting feature is that the median diurnal $O_3$ mixing ratios on weekends were consistently greater than on weekdays. Also, the

diurnal cycles of $O_3$ were inversely correlated with those for $NO_x$. The low $O_3$ mixing ratios in the mornings of weekdays can be explained by titration with high fresh emissions of NO from traffic, whereas the afternoon maximum is mainly due to production of $O_3$ through increased levels of photochemistry with VOCs. The diurnal cycle of odd oxygen ($O_x = O_3 + NO_2$) shown in Fig. 9 can be used to separate the contributions of titration and photochemistry with VOCs to $O_3$ mixing ratios. Titration does not increase the sum of $O_3$ and $NO_2$, whereas photochemistry with VOCs does. Therefore, variations in $O_x$

levels indicate that photochemistry with VOCs is important (Pollack et al., 2012). The average diurnal mixing ratios of $O_3$ and $O_x$ from NAPS measurements showing a maximum in the afternoon and being slightly higher on the weekends is also consistent with the average diurnal mixing ratios of HCHO showing a peak in the afternoon on weekends (Fig. 11). These diurnal results also suggest that the photochemistry with VOCs producing $NO_2$ and $O_3$ was important especially in the afternoon and on weekends. In addition, $O_x$ levels peaked in the afternoon, also consistent with diurnal cycles of sunlight

intensity (Fig. 2) which is a critical condition of photochemistry to produce $O_3$. Other VOCs besides HCHO that were emitted by traffic and other local sources may also have contributed to the photochemical production of $O_3$, but they were not quantified. Similar differences of $O_3$ and $O_x$ mixing ratios between weekdays and weekends were reported in the South Coast air basin (Pollack et al., 2012; Warneke et al., 2013). Temperature also affects $O_3$ production, but given the great variation of $O_3$ mixing ratio through the day, this was a secondary effect here based on box model calculations in the

temperature range of our study (Coates et al., 2016).



Time series of HCHO mixing ratios retrieved from the FTIR shown in Fig. 6 were between 0 and 5 ppb. The average diurnal cycle of HCHO during weekends shown in Fig. 11 did not correspond to the average diurnal cycles of either traffic volume or primary pollutant CO, but rather to the sunlight intensity (i.e., actinic radiation: see Fig. 2). This indicates that photochemistry of VOCs with oxidants was a dominant source (Stroud et al., 2016). The lifetime of HCHO in the
atmosphere is on the scale of hours to days depending on the levels of ambient oxidants (Seinfeld and Pandis, 2006). Stroud et al. (2016) reported on levels of HCHO in Toronto and Egbert, Ontario and source apportionment. Primary mobile emissions were found to contribute ~ 12% of HCHO in Toronto. Previous research also showed that both light-duty and heavy-duty vehicles emit HCHO (Grosjean et al., 2001).

The correlation between HCHO mixing ratio and ambient temperature was moderate ($r^2$ =0.42). HCHO levels were low for a
few weekdays (July 22-24) with lower temperature as compared to the four warmer days sampled on the weekends. Therefore, the difference between the average weekday and weekend diurnal cycles shown in Fig. 11 may be due in part to sample size, and it is possible that other local HCHO emission sources which depend on the temperature may also have contributed to the HCHO observed, especially in the afternoons on weekends.

GEM-MACH simulations of HCHO mixing ratio are always greater than the FTIR measurements (Fig. 6), but in the GEM-
MACH HCHO model species from the ADOM-II gas-phase chemistry mechanism is actually a lumped species that also includes isoprene oxidation products.

### 3.5     HCN

The mixing ratios of HCN retrieved from FTIR measurements were between 0 and 4 ppb (Fig. 6). HCN has severe adverse effects on human health, and chronic exposure to low cyanide can cause abnormal thyroid function and neurological
problems (El Ghawabi et al., 1975; Blanc et al., 1985; Banerjee et al., 1997; U.S. EPA, 2010). HCN has been reported previously in vehicle exhaust (Bradow and Stump, 1977; Keirns and Holt, 1978; Cadle et al., 1979; Urban and Garbe, 1979, 1980; Karlsson, 2004; Baum et al., 2007; Moussa et al. 2016). It may form over the catalytic converters in the vehicle emission control systems (Voorhoeve et al., 1975; Suárez and Löffler, 1986; Baum et al., 2007). A recent study for Toronto reported comparable HCN mixing ratio values (Moussa et al., 2016). These HCN measurements contribute to the few
studies reporting measurements of HCN mixing ratio beside a highway in urban ambient air.

### 3.6     CH₃OH

As shown in Fig. 6, mixing ratios of $CH_3OH$ from the FTIR were between 2 and 20 ppb most of the time, with some high spikes. Figure 12 presents the corresponding average weekday and weekend diurnal cycles of $CH_3OH$ for the study period. This plot shows the mixing ratio reached a peak (maximum of 20 ppb at 7:30) from 7:00 to 9:00 on weekdays whereas it was





generally flat on weekends, indicating that a large component of observed $CH_3OH$ may have come from traffic emissions. Observations of methanol associated with traffic have been reported in other studies. Rogers et al. (2006) reported $CH_3OH$ in the diluted pipeline exhaust of a mobile laboratory. $CH_3OH$ may also come from non-engine sources, such as windshield wiper fluid. Durant et al. (2010) measured gas and particle pollutants near Interstate 93 in Massachusetts. They reported $CH_3OH$ was above 20 ppb at 7:20 50 m downwind of the highway, possibly with contributions from some other local

sources.

### 3.7    Estimation of emission factors

To evaluate the feasibility of using measurement data from this study to estimate emission rates, we picked measurements for three days from this study to use as inputs to a backward Lagrangian stochastic dispersion model (WindTrax, http://www.thunderbeachscientific.com/). The following were included as the inputs: CO mixing ratio from the FTIR;

background mixing ratio of CO; winds and temperature from the NAPS trailer; and atmospheric stability ($u_*$ and L) from the scintillometer. The surface roughness ($z_0$) was set to 0.15 m, the maximum allowed by WindTrax. The defined section of Highway 401, which was assumed to be the only CO source in the footprint, is about 1870 m long and 110 m wide. The FTIR path is roughly in the centre of the defined section. WindTrax was then used to estimate the emission rate of CO from this defined section. In the model, 50000 virtual particle trajectories were calculated upwind of the FTIR path with the given

meteorological conditions (wind direction and temperature) and surface-layer turbulence ($u_*$ and L), to determine what fraction of trajectories originated from the designated source area.

Over three days, July 22, 28, and 29, CO emission rate estimates were calculated by the WindTrax with one-minute resolution for ten-minute periods, and the average estimates over those ten-minute periods were calculated and shown as the markers in Figure 13. The constant background used in WindTrax was 265 ppb, which was determined from the CO

intercept of the linear regression analysis of $NH_3$ with CO (see Section 3.3). The mixing ratio of changing background used in WindTrax was determined using a more dynamic definition of background based on wind direction. When the wind was from the south, the background was chosen as the NAPS measurement for July 28 and 29. In the morning on July 28 when the wind was from the northwest, the background was chosen as 415 ppb, the minimum mixing ratio of that morning measured by the FTIR. When the wind direction varied greatly, the background value of the previous hour was chosen. On

July 22, the wind was consistently from the northwest, and the minimum of 329 ppb over the whole day from the FTIR was chosen as the changing background.

Liu and Frey (2015) reported vehicular CO emission factors ranging from 0.003 to 5.1 g km$^{-1}$ with an average value of 0.62 g km$^{-1}$ based on empirical data measured between 2008 and 2013 in the Raleigh and Research Triangle Park area (North Carolina, U.S.) for 100 vehicles with a range of model years and accumulated mileage. They also reported simulated CO

emission factors ranging from 0.004 to 6.87 g km$^{-1}$ with an average value of 1.99 g km$^{-1}$ by using the U.S. EPA's Motor



Vehicle Emission Simulator (MOVES) emission factor model. Moussa et al. (2016) reported that emission factors of CO measured from several gasoline light duty vehicles with different driving cycles ranged from 0.1 to 3.0 g km$^{-1}$ with an average value of 0.9 g km$^{-1}$

Using emission factors from the MOVES model, traffic volume estimates, and the width of Highway 401 at the site, a "bottom-up" estimate of the emission rate was calculated and compared to the dispersion model results (Fig. 13). There is a good agreement amongst the different approaches. On July 22, the emission rate estimates from the WindTrax are close to the MOVES results. The MOVES estimates show a sharp drop between 14:00 and 15:30 due to decrease of traffic volume to 14 % of at the value at 13:30. During this time, the emission rate estimates from the WindTrax do not fluctuate much. This result suggests that vehicle numbers passing by a fixed point may not be the best indicator of emissions since they do not

account for the traffic speed: an extreme example would be a traffic jam with zero traffic flow but nonzero emissions. On July 28 and 29, the estimates from the WindTrax are greater than the MOVES estimates. The difference between the emission rates estimated by using the WindTrax with changing background and by using 0.9 g km$^{-1}$ and 3.0 g km$^{-1}$ from Moussa et al. (2016) is in the range of -140 % to 460 %, and -110 % to 70 %, respectively. These results suggest that WindTrax dispersion calculation results based on CO mixing ratio measurements from the FTIR and in-situ

micrometeorology are well within the range of estimates based on traffic volume and emission factors of various vehicles.

The input background mixing ratio assumed at each hour influences the emission rate estimates. Especially during the night around 0:00 to 6:00 when the wind was from north, it is difficult to determine the background mixing ratios of CO, since no measurements were available upwind. Both emission rate estimates with constant background and changing background were calculated to check the sensitivity of background mixing ratio on the emission rate estimates by the WindTrax. Our

changing-background approach should be closer to reality, since conditions around the highway do change with time. Figure 13 shows that both estimates from the WindTrax using changing background and constant background agree in general with the bottom-up estimates, except for the period of the morning on July 29 when WindTrax estimates with the constant background are greater than the other estimates. The wind was from the south during this period and the assumed constant background of 265 ppb was significantly lower than the NAPS measurements, resulting in an overestimate by the WindTrax.

Beside the uncertainties in background mixing ratio, the variations in emission rate estimates are most likely due to changes in wind direction over short periods. When the wind direction changed quickly, the input wind direction used by the WindTrax may not be representative and hence may bias the calculated emission rate.

Emission rates of other primary pollutants from traffic can be determined by using the concentration ratios of these pollutants to CO and emission rate estimates of CO, as mentioned in Section 3.2.1. We found that [NH$_3$] (as discussed in

Section 3.3), [CH$_3$OH], and early morning periods of [NO] had linear relationships with [CO]. With these ratios and emission rate estimates of CO obtained from the WindTrax (Figure 13), the ten-minute average emission rate estimates of



NH$_3$, NO, and CH$_3$OH were calculated. The minimum-to-maximum range and the average of these ten-minute average estimates are shown in Table 2.

The inputs of u$_*$ and L from the scintillometer measurements also contain uncertainties. A realistic estimate of the
uncertainty of u$_*$ is ± 20 % to 30 % (Andreas, 1992). We conducted a sensitivity study by varying u$_*$ from 0.7 to 1.3 u$_{*(obs)}$
and L from 2.20 to 0.34 L$_{(obs)}$ corresponding to change of u$_*$ while keeping heat flux fixed, which resulted in emission rates
from 0.69 to 1.37 times the original emission rate estimates. We also investigated the sensitivity of heat flux on emission
rates estimates by varying L from 0.5 to 2L$_{(obs)}$ with fixed u$_{*(obs)}$, which resulted in emission rates estimates are in range of
0.70 to 1.45 times of original emission rate estimates. Therefore, even with conservative uncertainty estimates about
surface-layer stability, the calculated emission rates from the WindTrax are still within 45 % of the bottom-up estimates.

WindTrax limits z$_0$ to a maximum value of 0.15 m. However, the z$_0$ of the actual measurement site over the highway is
around 0.6 m based on urban-scale meteorological model results for Toronto (Leroyer et al., 2016). This difference between
the actual z$_0$ and the input z$_0$ used by WindTrax likely also contributes to the uncertainty of the emission rate estimates.

Better emissions-rate estimates might also be obtained if traffic information included vehicle types and vehicle speed. Speed
and speed variation of different vehicle types are known to affect emission rates of CO, hydrocarbon and NO$_x$ (Zhang et al.,
2011), but consideration of these factors would have required more sophisticated traffic quantification.

## 4.    Summary and Conclusions

This study demonstrated the utility of combining long-path FTIR spectroscopy with micrometeorological measurements to
identify and quantify pollutants emitted by moving traffic, and to calculate emission rates in a representative real-world
setting. We retrieved mixing ratios of eight air pollutants over Highway 401 in Toronto, Canada. Traffic emissions were
shown to contribute quantifiable levels of NH$_3$, HCN, HCHO, and CH$_3$OH to the urban mix of pollutants. Of particular
interest was the quantification of species such as HCN, a toxic pollutant with severe health implications, and NH$_3$, which
may be gaining in importance due to the increasing use of catalytic converters which reduce vehicular NO$_x$ emissions. Very
few ambient data sets on these species from traffic-dominated environments are available in the published literature, and the
methods described here can fill a significant gap.

Differences between weekdays and weekends in the average diurnal cycles of some of the pollutants mixing ratios (CO,
NO$_x$, O$_3$, NH$_3$, and HCHO) were observed. The biggest differences are that on weekdays, the mixing ratios of primary
pollutants from traffic, such as CO and NH$_3$, showed an obvious peak in the early morning around 6:00 to 9:00,
corresponding to the sharp increase of traffic volume during morning rush hour, while on weekends, mixing ratios varied



less throughout the day and no obvious peaks in the early morning were observed. Combined FTIR analysis and turbulence results clearly elucidated the role of turbulence in the build-up and dispersion of traffic emissions.

A comparison of the path-averaged FTIR with single-point NAPS measurements showed general agreement of the variations in mixing ratio, but also showed differences due to the difference in measurement footprint. This comparison also uncovered some issues with offsets and amplitude differences between the FTIR and in-situ analyzers that are likely due to pervasive

$H_2O$ interference across the FTIR spectrum, especially for NO and $NO_2$.

The modelled pollutant concentrations at the study site from a high-resolution version of the GEM-MACH air quality model agreed well in general with the measurements, especially for CO, $O_3$, and $NH_3$. Given that the version of GEM-MACH considered here employed 2.5 km by 2.5 km grid cells, model results and measurement results are not expected to directly comparable for all wind regimes, but reasonable correlations were observed.

Lastly, by combining mixing ratio with micrometeorological measurements and a simple dispersion model, we demonstrated the calculation of real-world, spatially representative vehicular emission rates using CO as an example, and derived emission rates of $NH_3$, NO and $CH_3OH$.

## 5. Acknowledgements

We thank Andrew Sheppard, Andrew Elford, Roman Tiuliugenev, Raymon Atienza, and Rajananth Santhaneswaran

(Environment and Climate Change Canada, ECCC) for their technical support, Richard Mittermeier (ECCC) for his help on the FTIR measurements and suggestions on FTIR analysis, the NAPS program (ECCC) for providing instruments to the NAPS trailer, Peter Maas (Bruker) for his suggestions on measuring and analyzing results using the OPS software, Aldona Wiacek and Li Li (Saint Mary's University) and David Griffith (University of Wollongong, Australia) for their suggestions on retrieving concentrations from the FTIR, Terry Gillis (Pine Point Arena) for accommodating the retroreflector and LED

array; Matthew Tuen (Ontario Ministry of Transportation) for providing the traffic volume data, Peter Taylor at the York University for providing meteorological data, Tak Chan and John Liggio (ECCC) for their comments on vehicle emissions, Sumi Wren and Jeff Brook for sharing results on their measurements of pollutants in urban Toronto, Andrea Darlington (ECCC) for her help on Igor program functions, and Chris Sioris (ECCC) for his review of the manuscript. We also acknowledge the developers of the OpenAir air quality analysis package for this remarkable tool (Carslaw and Ropkins,

2012; Carslaw, 2015).

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





## Tables

**Table 1. Regions of long-path FTIR spectra used to retrieve mixing ratios of target gases in this study.**

| Gases of Interest | Spectral Region Fitted ($cm^{-1}$) | Interference Gases Fitted | Correlation Threshold (r)[a] | Detection Limit (ppb)[b] |
|---|---|---|---|---|
| CO (Carbon monoxide) | 2142-2241 | $H_2O$, $CO_2$, $N_2O$ | 0.7 | 0.7 |
| $CO_2$ (Carbon dioxide) | 2224-2255 | $H_2O$, $N_2O$, CO | 0.97 | |
| $CH_4$ (Methane) | 2904-3024 | $H_2O$ | 0.95 | 0.7 |
| $O_3$ (Ozone) | 1040-1065 | $H_2O$, $NH_3$, $CH_3OH$, benzene, HCHO | 0.7 | 4.4 |
| NO (Nitrogen oxide) | 1893-1913 | $H_2O$ | 0.1 | 5.8 |
| $NO_2$ (Nitrogen dioxide) | 1595-1607 | $H_2O$, $NH_3$, $CH_3OH$ | 0 | 7.3 |
| $SO_2$ (sulfur dioxide) | 2465-2550 | $H_2O$, $N_2O$ | 0.5 | 1.5 |
| $CH_3OH$ (Methanol) | 980-1080 | $H_2O$, $NH_3$, $O_3$ | 0.7 | 0.7 |
| $NH_3$ (Ammonia) | 910-990 | $H_2O$ | 0.7 | 0.7 |
| HCN (Hydrogen cyanide) | 710-717 | $H_2O$, $N_2O$, $CO_2$, $C_2H_2$, $NH_3$, $NO_2$ | 0.3 | 2.9 |
| HCHO (Formaldehyde) | 2740-2840 | $H_2O$, $CO_2$, $CH_4$ | 0.3 | 1.5 |
| $N_2O$ (Nitrous oxide) | 2198-2223 | $H_2O$, $CO_2$, CO | 0.97 | 0.5 |
| $CH_3(CO)CH_3$ Acetone | 870-940 | $H_2O$, $NH_3$, $C_2H_6$, $C_2H_4$ | 0.3 | 2.2 |
| $C_2H_2$ (Acetylene) | 680-780 | $H_2O$, $CO_2$ | 0.3 | 0.6 |
| $C_2H_6$ (Ethane) | 800-850 | $H_2O$, $CO_2$ | 0 | 1.4 |
| $C_3H_8$ (Propane) | 2860-2975 | $H_2O$, $CH_4$, $C_2H_6$, HCHO, $NO_2$ | 0 | 0.7 |

[a] Correlation thresholds are inputs for OPS used when retrieving the mixing ratios from FTIR spectra. When the correlation
between the measured spectrum and reference spectrum in that spectral range is below this threshold, that pollutant is not
"identified" and the mixing ratio will be reported as zero.
[b] Detection limit is $3\sigma$ of the noise for measurements with a retroreflector distance of 310 m according to Bruker.






**Table 2. Pollutant emission rates**

| Pollutant | [pollutant]/ [CO] (ppbv/ppbv) | Emission rates (g m$^{-2}$ h$^{-1}$) | Emission factors (average) (g km$^{-1}$) | Emission factors (g km$^{-1}$) previously reported |
|---|---|---|---|---|
| CO | 1 | 0-0.90 | 0-6.97 (2.6) | 0.004-6.84 (Liu and Frey 2015) 0.1-3.0 (Moussa et al. 2016) |
| NH$_3$ | [a]0.026 | 0-0.01 | 0-0.11 (0.04) | 0-0.11 (Durbin et al. 2002) 0-0.144 (Huai et al. 2003) 0-0.26 (Livingston et al. 2009) 0.004-0.062 (Suarez-Bertoa et al. 2014) |
| NO | [b]0.128 | 0-0.12 | 0-0.96 (0.36) | 0.008-1.26 Frey et al. (2003) |
| [c]CH$_3$OH | [c]0.051 | 0-0.05 | 0-0.41 (0.15) | N/A |


[a] The ratio was found as the slope of the linear fit over the entire study period.
[b]The ratio was found as the average slope of the linear fits for the three early morning periods on July 22, 28 and 29.
[c]The ratio was found as the slope of the linear fit for July 28 and 29.



**Figures**

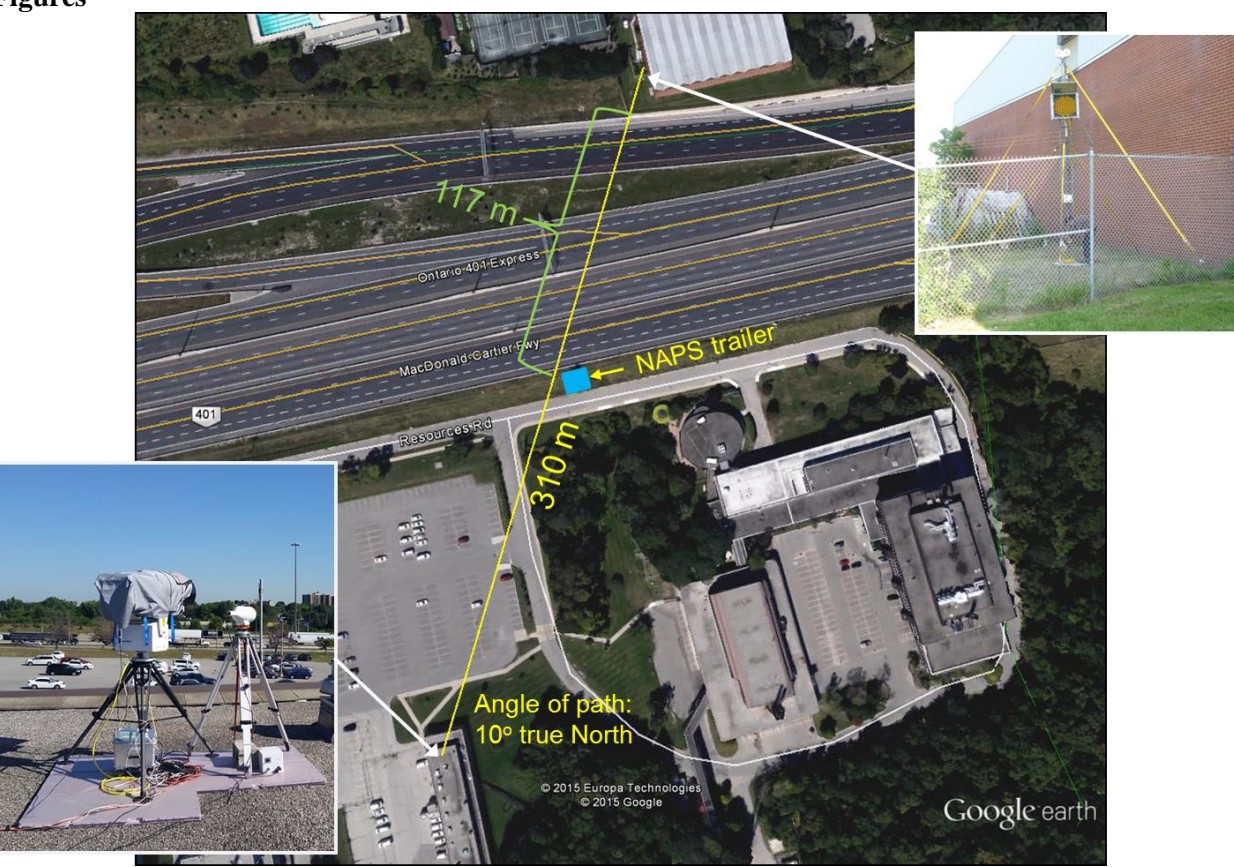


**Figure 1: Setup of the FTIR, scintillometer (see Section 2.2), and the NAPS trailer near Highway 401.**






**Figure 2: Average diurnal cycles (one-hour averages) of z/L and u$_*$ (top), CO mixing ratio from the FTIR (middle), as well as sensible heat flux H and downwelling shortwave radiation (bottom) for 16-31 July 2015.  Lines represent the medians, and the shaded regions are the interquartile ranges for z/L, CO, and H.**



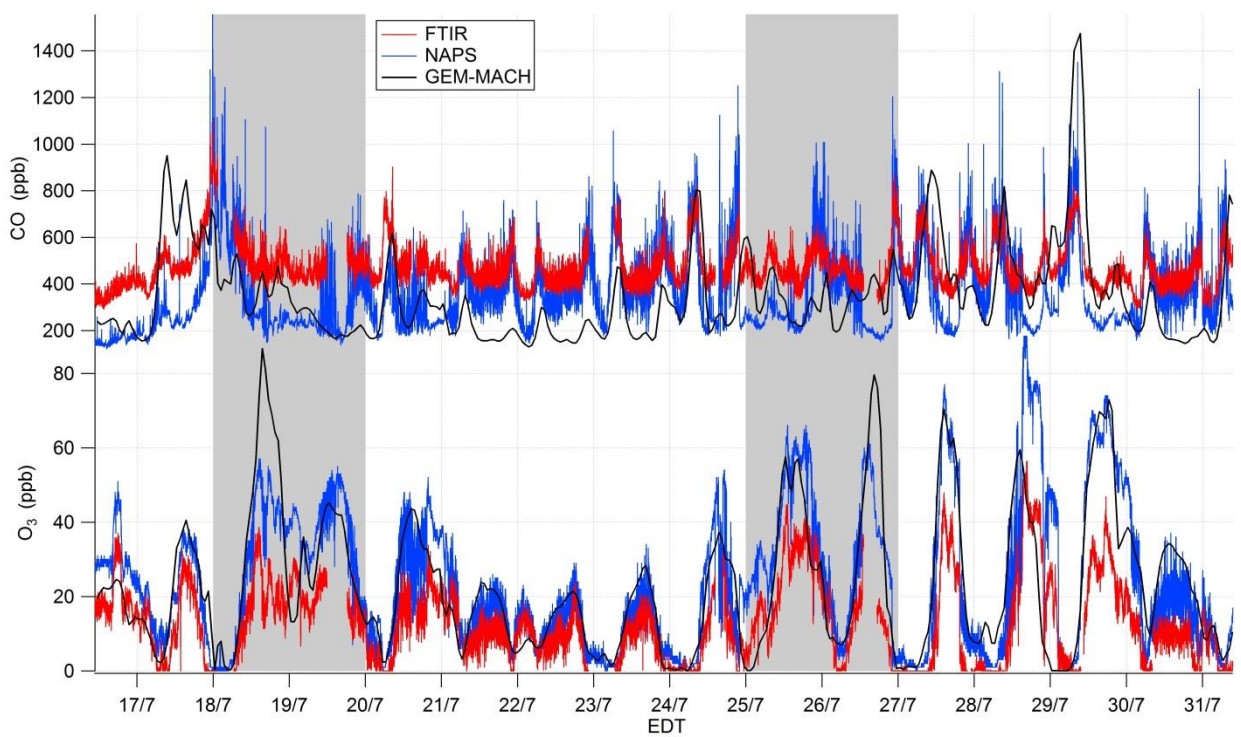

**Figure 3: Time series of mixing ratios of CO (top) and O₃ (bottom) for the full study period. The red traces are mixing ratios retrieved from the FTIR spectra. The blue traces are measurements from the NAPS station. The black traces are output from GEM-MACH. Grey shaded areas highlight the weekend periods.**



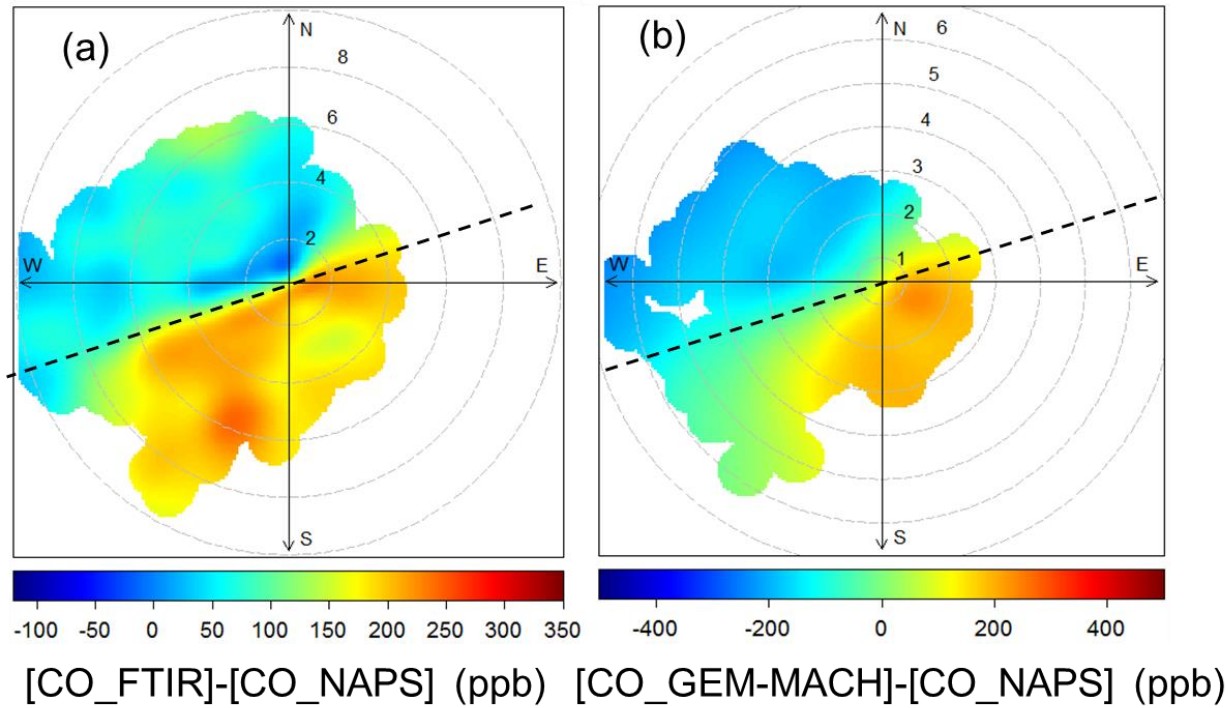

[CO_FTIR]-[CO_NAPS] (ppb)  [CO_GEM-MACH]-[CO_NAPS] (ppb)


**Figure 4: Polar plots of CO mixing-ratio difference between measurements from the FTIR and NAPS (a), between GEM-MACH output and NAPS measurements (b). Azimuth angle represents wind direction (meteorological convention: 0° = wind from north, 90° = wind from east, etc.), and radius indicates wind speed (m s⁻¹). The color shows the CO mixing ratio difference. The center corresponds to the location of the NAPS trailer. The black dashed line shows the orientation of the highway: above this line, the**

**wind came from the highway towards the trailer.**







**Figure 5: Average weekday and weekend diurnal cycles of mixing ratio of NH₃ (top) and CO (middle) from the FTIR and traffic volume (bottom) for the 16-day study period. Blue solid lines are medians, and the shaded areas show the interquartile ranges for weekdays; black dashed lines are the medians for weekends. The brown solid line is an estimation of NH₃ levels associated with traffic emissions on weekdays; the brown dashed line corresponds to weekends.**






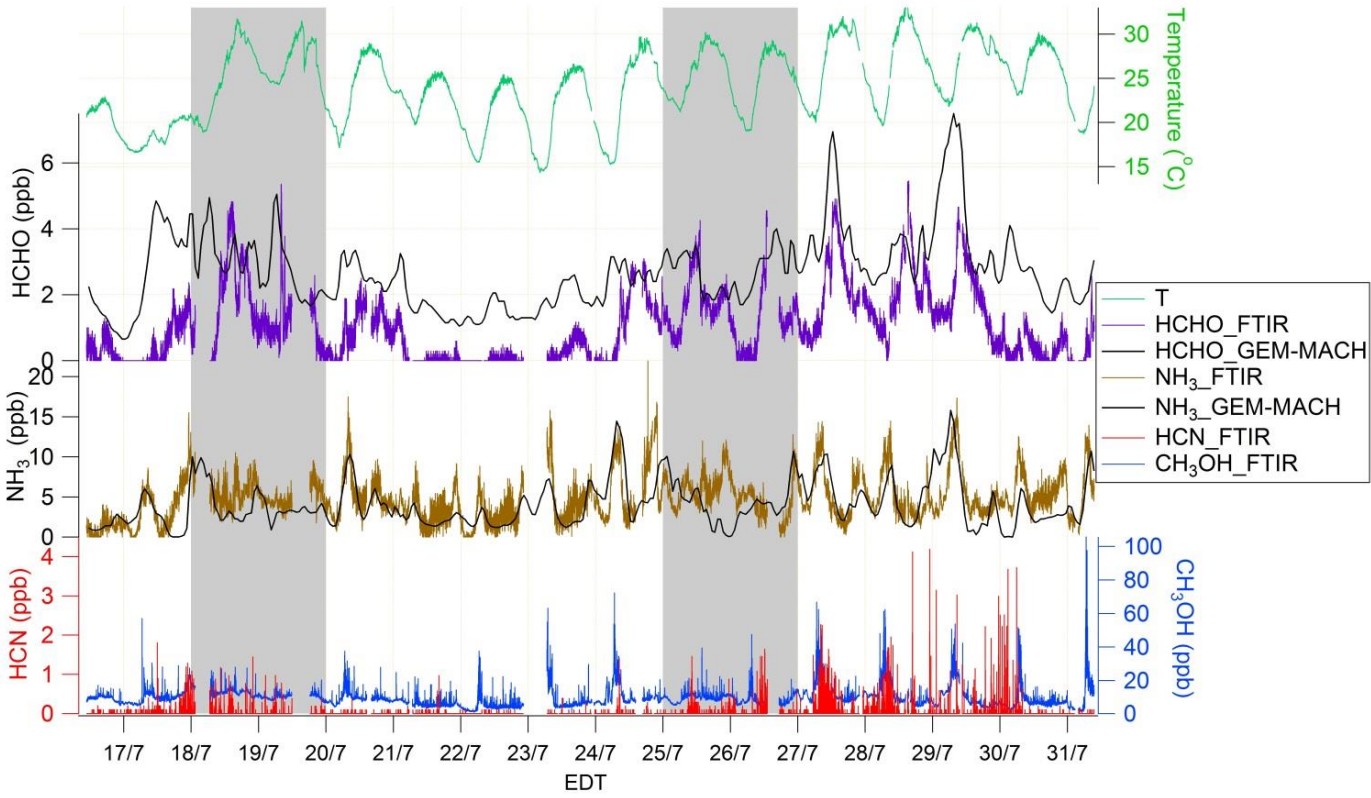

**Figure 6: Time series of ambient temperature, mixing ratio of HCHO, NH₃, HCN, and CH₃OH for the 16-day study period. Grey**
**shaded areas indicate the weekend periods.**





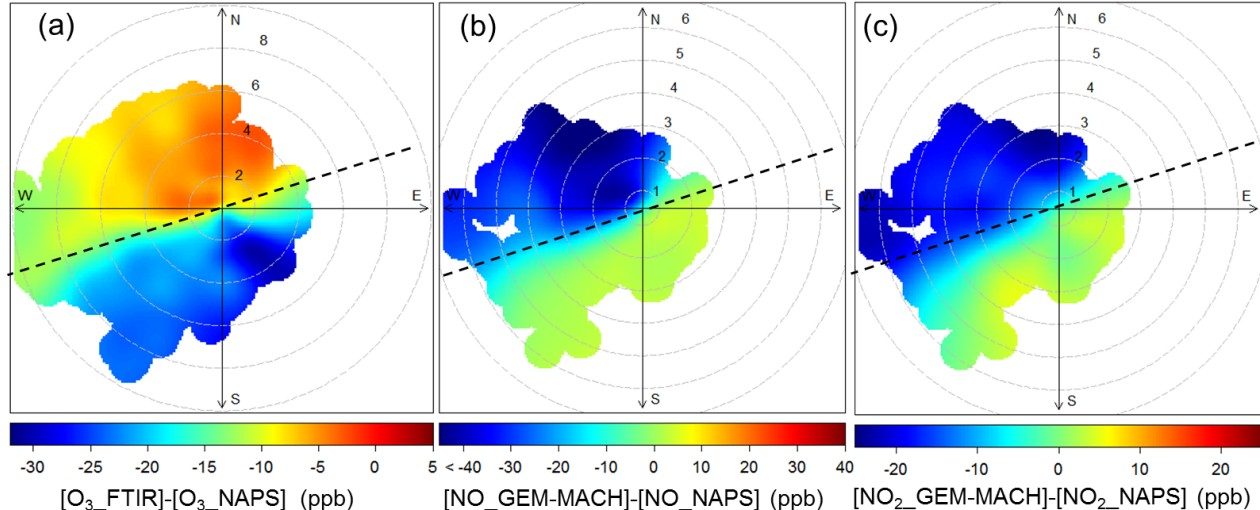

**Figure 7: Polar plots of O$_3$ mixing ratio difference between measurements from the FTIR and the NAPS (a); mixing-ratio difference between results from GEM-MACH predictions and hourly averaged measurements from NAPS for NO (b) and NO$_2$ (c). Azimuth angle represents wind direction (meteorological convention), and radius indicates wind speed (m s$^{-1}$). The center of each plot corresponds to the location of the NAPS trailer. The black dashed line shows the orientation of the highway: above this line, the wind came from the highway towards the trailer.**



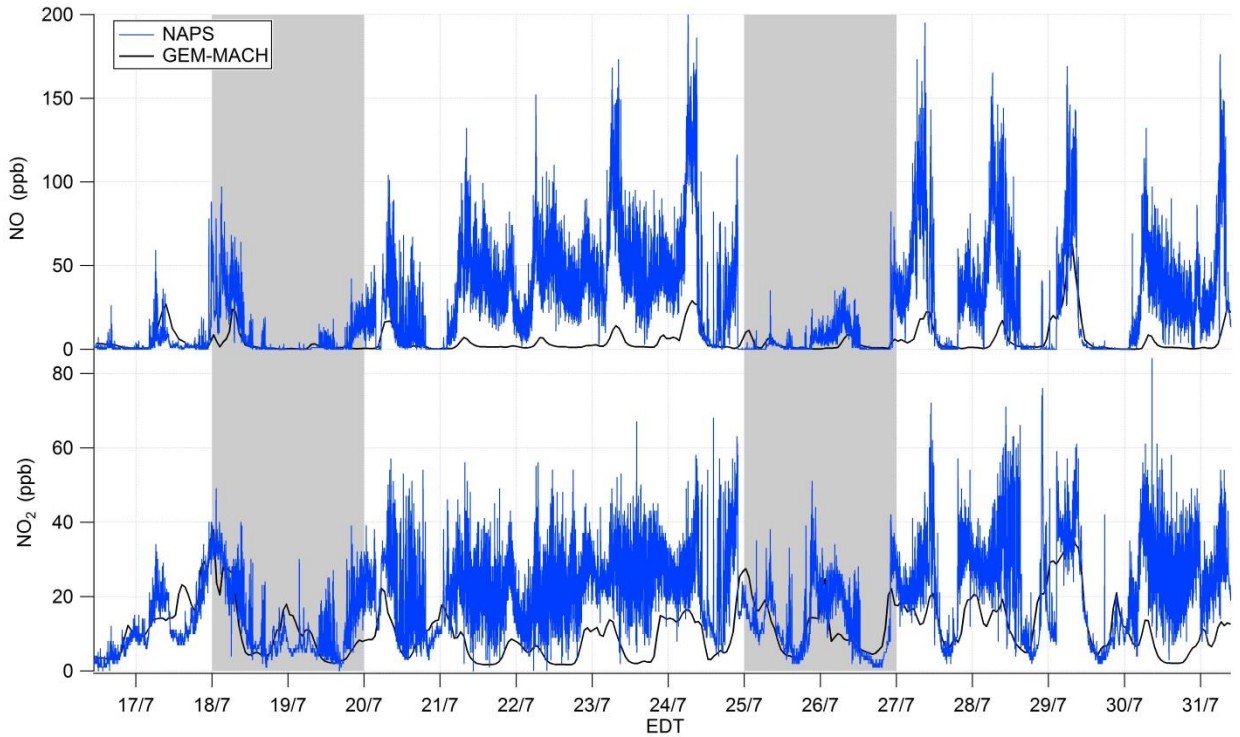

**Figure 8: Time series of mixing ratios of NO (top) and NO₂ (bottom). Grey shaded areas indicate the weekend periods.**





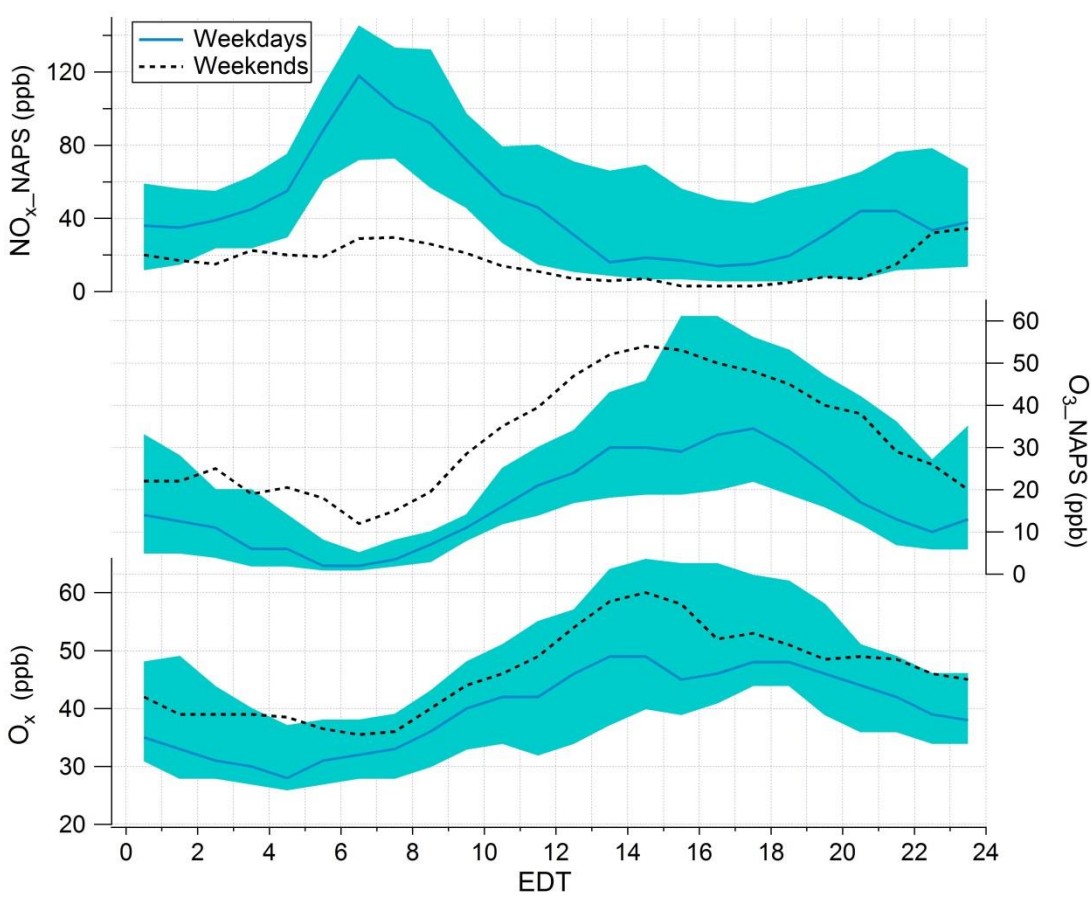

**Figure 9: Average weekday and weekend diurnal cycles of mixing ratios of $NO_x$ (top), $O_3$ (middle), and $O_x$ (bottom) from the NAPS for the 16-day study period. Solid green lines are medians and the shaded areas are the interquartile ranges on weekdays; dashed black lines are medians on weekends.**





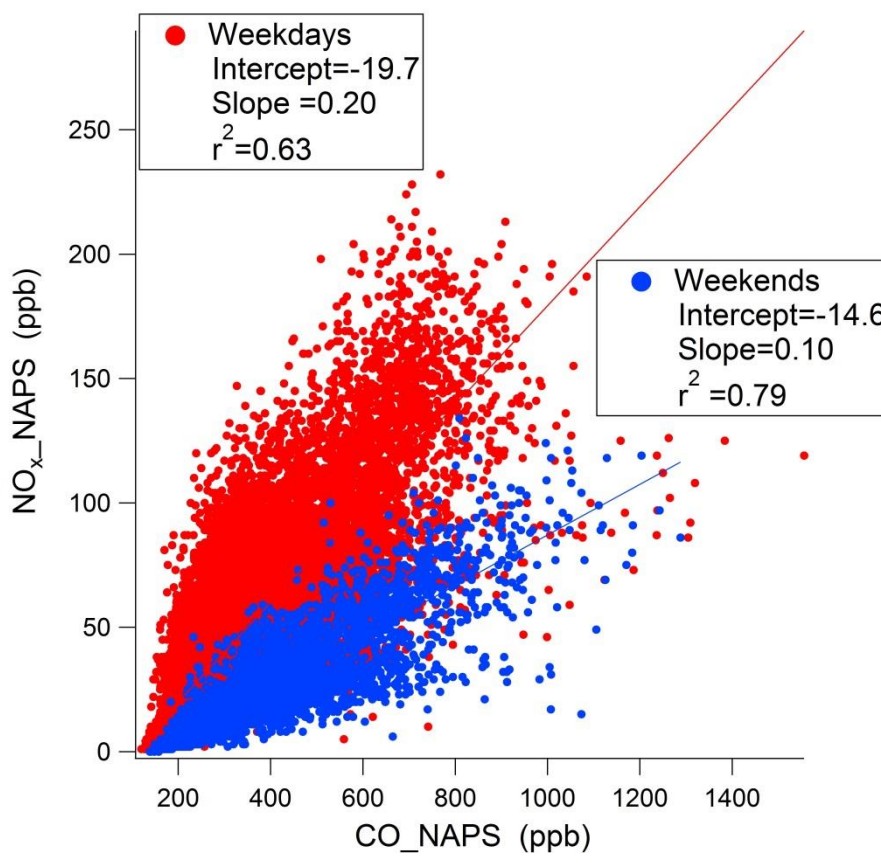

**Figure 10: Scatterplot of $NO_x$ vs. CO mixing ratios from the NAPS on weekdays (red) and weekends (blue). Lines are the linear**
**regression results for weekdays (red) and weekends (blue).**





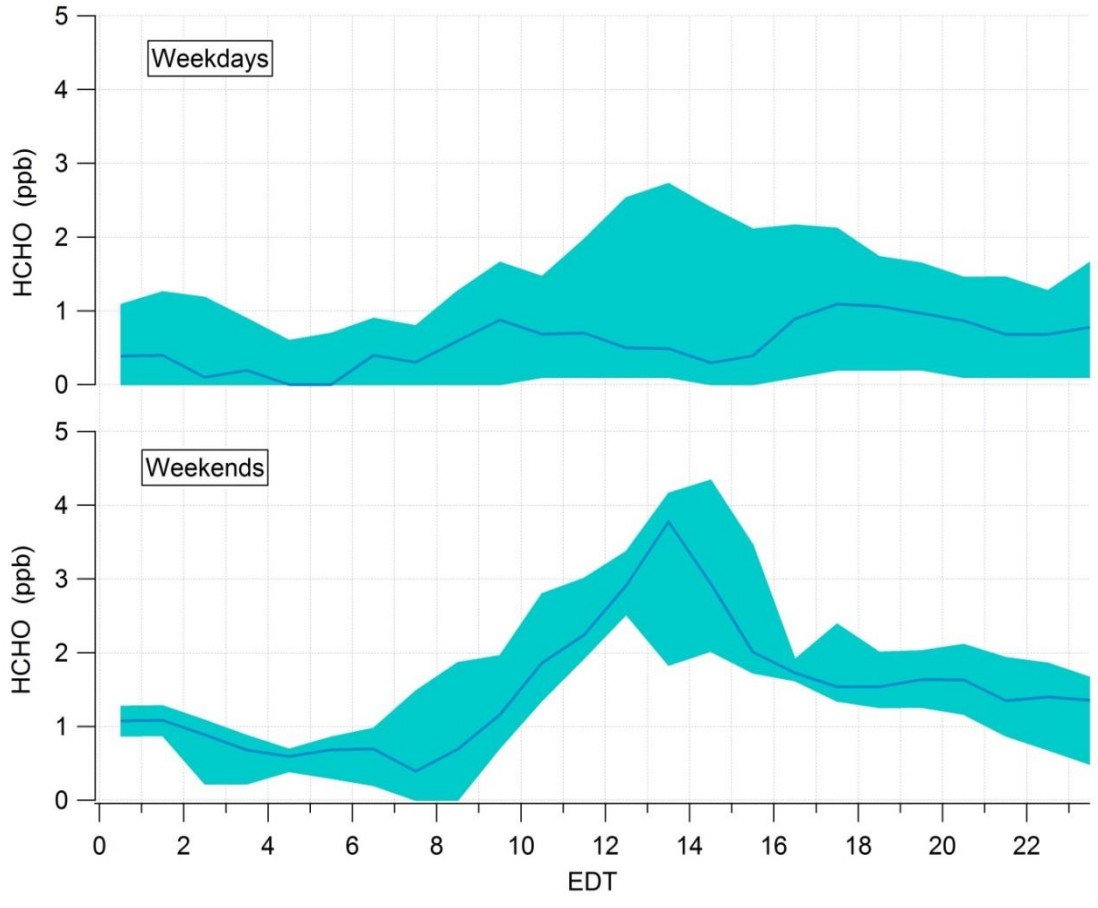

**Figure 11: Average diurnal cycles of HCHO on weekdays (top), and on weekends (bottom) for the 16-day study period. Solid green lines are medians, and the shaded areas are the interquartile ranges.**





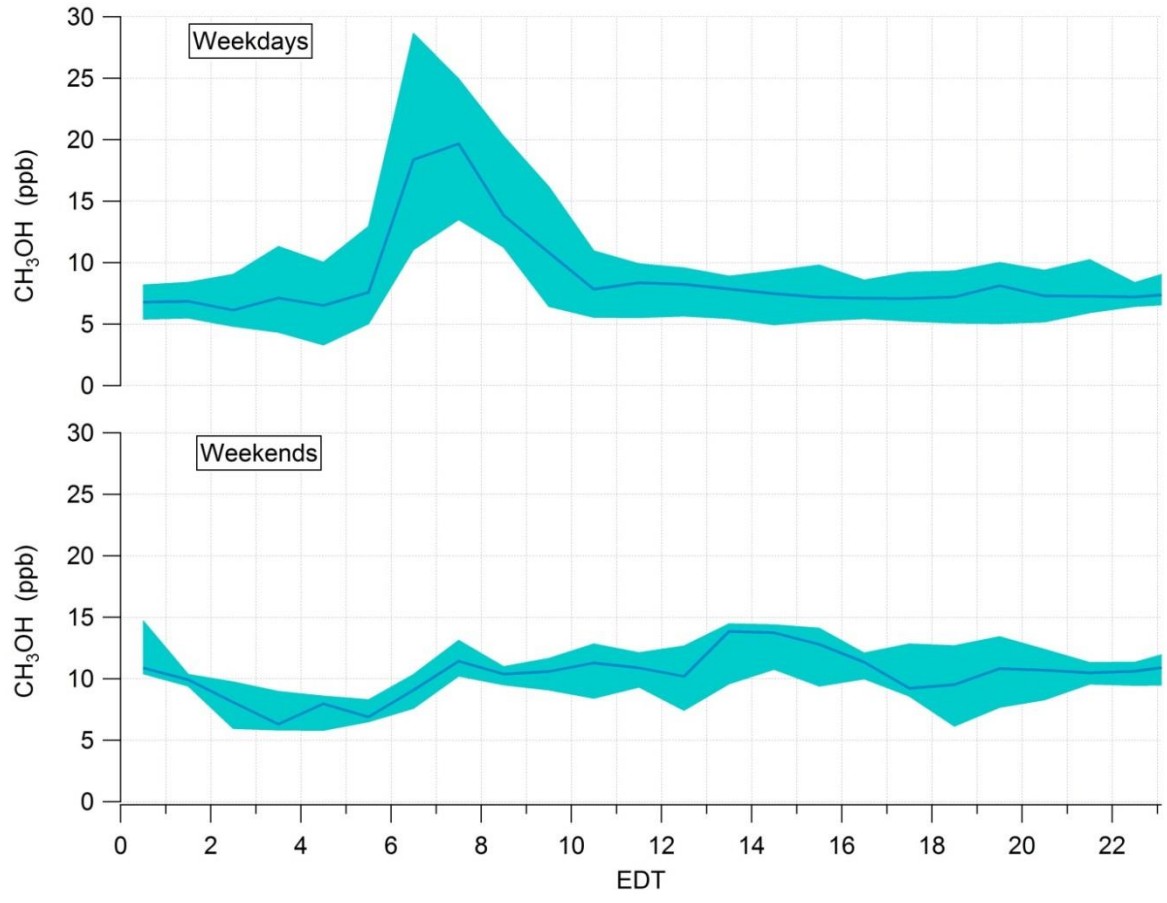

**Figure 12: Average diurnal cycles of CH$_3$OH mixing ratio on weekdays (top) and weekends (bottom) for the 16-day study period. Solid green lines are medians, and the shaded areas are the interquartile ranges.**






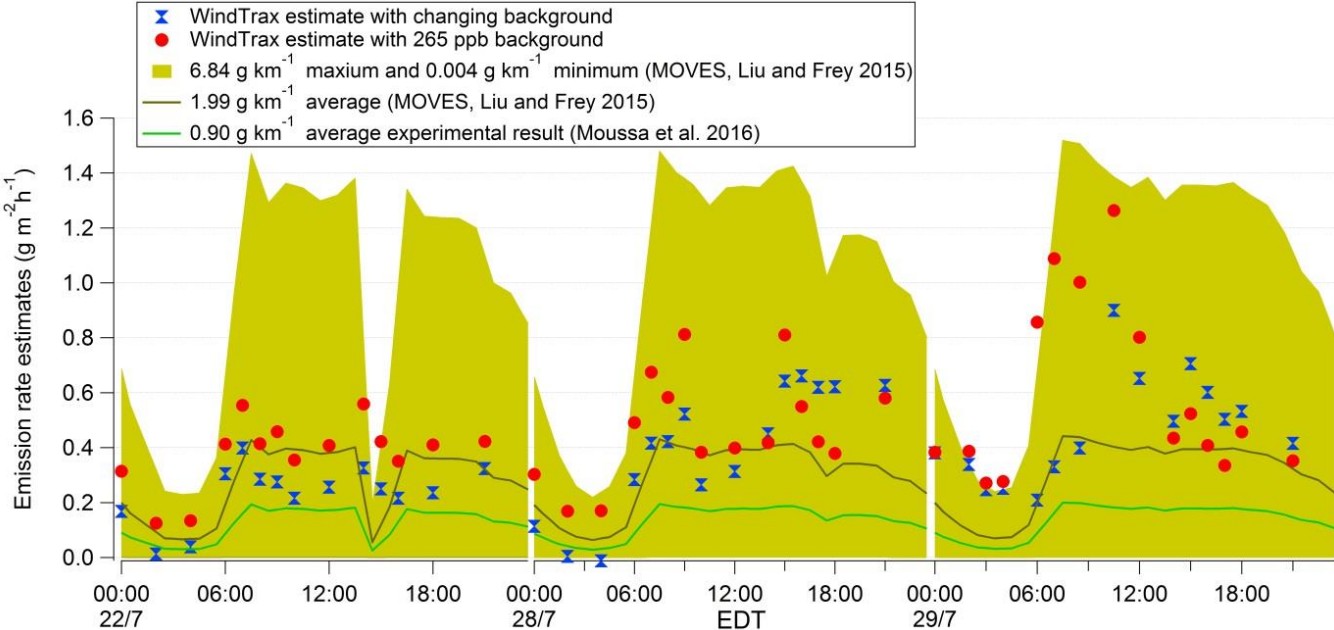


**Figure 13: CO emission rate estimates over three days. Red dots are CO emission rates simulated by WindTrax using CO mixing ratios from the FTIR and a constant CO background of 265 ppb (see the text). Blue markers are CO emission rates simulated by the WindTrax using changing CO background values. The brown line is the CO emission rate estimated by using traffic volume estimates and emission factors from the average MOVES results in Liu and Frey (2015); the brown shade is the range of CO**
**emission rates estimates obtained by using the maximum and minimum CO emission factor results from MOVES in Liu and Frey (2015). The green line is the CO emission rate simulated by using traffic volume estimates and the average CO emission factor from Moussa et al. (2016).**
