# Peer review of "Long-path measurements of pollutants and micrometeorology over Highway 401 in Toronto"

_Atmospheric Chemistry and Physics, 2017_

## Referee Comment (RC1) · Anonymous Referee #1 · 28 May 2017

This paper provides a significant contribution to the study of air quality within urban environments. The combination of open-path FTIR measurements with in-situ measurements and scintillometer measurements is interesting and novel. The paper is well written, clear and scientifically sound. I enjoyed reading it! In my opinion it should be accepted for publication after some minor comments are addressed as outlined below:

1. Line 113-line 116: this needs slight elaboration – the stray light problem should be very minor and only from internal reflections (since the light is modulated before being sent across the open-path. This means ambient scattered radiation will not be modulated and thus not detected by the instrument).

2. The MALT program (which you say that your analysis is based on) can calculate the

reference spectra directly from the HITRAN database using the ambient temperature measured at the same time as the spectra were recorded. Does the Bruker software really not allow this? This may only introduce uncertainties of 10% but it is an unnecessary added uncertainty, since you have accurate temperature measurements available.

3. There are other uncertainties inherent to open-path FTIR measurements (like those that come from the HITRAN database and fitting errors). These are not mentioned in the text but should at least be referred to as existing even if a full uncertainty analyses is not given.

4. Section 2.2 – at what time resolution are these calculations made?

5. Line 212 – Isn't an estimate using WindTrax and CO mole fractions a "top-down" estimate? You then compare it to one based on traffic volumes – isn't this one "bottom-up"?

6. Line 247: do they generally agree? The level of agreement is not quantified. From looking at the time-series the variations certainly seem to be well captured, but it would be good to give a correlation coefficient for this.

7. In fact some basic statistics for the model's skill level would improve the manuscript. The authors have used the "open-air" package and this has some great tools for quick evaluation of a model's performance against observations.

8. Page 10: reading this discussion about model to measurement spatial differences begs the question as to why a comparison of model to open-path FTIR is not shown. This will suffer similar problems but should be much less than the in-situ observations.

9. Line 297. Are the traffic volumes similar on the weekends? This is surprising and you have not actually stated that clearly before. Can you clarify?

10. Line 323: is it worth showing the correlation plot at least in the supplementary data?

11. Line 332: Are you also assuming that traffic is the only source of CO emissions above background?

12. Line 438: Why only 3 days? I assume that these are the best steady wind conditions? Whatever the reason for the choice of these days, it should be stated briefly in the text.

13. Line 529: "reasonable" correlations observed...You need to provide some actual statistics to back this up somewhere in the manuscript.

Other minor points

1. Consider changing "mixing ratio" to "mole fraction" throughout, as I believe this is now the preferred terminology.

2. Line110: the "fraction" of the path is not actually a fraction but a distance, consider rephrasing.

---

## Referee Comment (RC2) · Anonymous Referee #2 · 4 Jun 2017

Review of Manuscript "Long-path measurements of pollutants and micrometeorology over Highway 401 in Toronto," by Y. You, R.M. Staebler, S.G. Moussa, Y. Su, T. Munoz, C. Stroud, J. Zhang and M.D. Moran.

The paper presents a study by You et al. using open-path FTIR measurements to study traffic emissions across a large motorway located in an urban environment. These FTIR data are combined with a series of micrometeorological instruments to help identify / quantify a range of various air pollutants. Results are compared with in situ observations at a roadside monitoring station and also modelled using an air quality forecast model. Mixing ratios of $NH_3$, $HCN$ and $CO$ and other gases and vapours

were observed, with elevated mixing ratios for NH3. Diurnal and weekly variations are observed and discussed. The paper is generally well written and the English is good. Certain aspects of the paper are new (monitoring near such a large motorway) and may be worthy of publication, but in many regards the work lacks novelty. The greatest shortcoming of the paper, however, is that there appear to be serious flaws in the data or data analysis, particularly the FTIR spectral analysis. Indeed, it appears that the FTIR data were not analysed at all, but simply taken "as is" from the system, and the IR data appear to have large systematic errors. As a consequence we recommend that the paper be either rejected or reconsidered only after major revision whereby the authors completely reanalyse the IR data (or actually analyse for the first time!) using a more sophisticated and interactive approach. That the IR data or data analysis are fundamentally flawed will be discussed below.

With regards to novelty of the study, the crux of the paper is to collect FTIR measurements over an urban highway. Such works have been previously reported – see for example work by Bishop et al., Stremme et al., Colman et al., Grutter et al. and Chaney et al, as well as by other analytical methods (for example, the seminal paper by Stedman et al., the tunnel study by Popa et al and the newly released paper by Haugen et al.). That is to say, there exists significant literature regarding traffic emissions, and the present authors need to cite more of these studies, and also need to cite other studies that use open-path FTIR to measure similar compounds from other sources (for example, the several papers by Griffith et al., or the FTIR biomass burning studies using FTIR by Yokelson et al or the volcanic emission studies by Oppenheimer et al.). To add a more unique aspect to their study, we suggest that the authors i) emphasize more the aspect of monitoring near such a large motorway (e.g. perhaps there are atmospheric reactions / products only seen due to higher order rate constants requiring very high CO or O3 levels?), and ii) present a more detailed analysis of the instrumental comparison, i.e. the "shoot-out" between the measurements collected from the FTIR and from the NAPS. However, the authors must beware that such comparison studies are tricky because even if the instruments do in fact agree (within the error bars), then

it could mean that both are right or it still could potentially also mean that both are wrong as they have the same systemic errors. If the instruments do not agree then it could mean one instrument is right, the other is wrong, or even both are wrong! In the present case where the instrument values do not agree, the authors have largely chalked it up to a combination of a) the NAPS measuring as point source while the FTIR is measuring sources within a path, and b) wind variations dispersing the auto exhausts in different directions. Such generalities may be true, but do not provide much insight, except for raising the question as to whether the comparison as an "apples to oranges" point source v. path averaged measurement. If indeed the comparison is not valid, then what was the significance of doing the comparison, i.e. making the measurements? It seems the whole point of the study is to provide definitive gas phase measurements for certain species. However, at the end of the paper, it is unclear what are the reliable mixing ratios determined during the study. Are the FTIR values correct? Are the NAPS data correct? These author suggest that both instruments are correct and that the difference in the mixing ratios is due only to wind inhomogeneity and/or different magnitudes of plume mixing. This is not really supported by the data, and does not provide much insight.

We believe (at least some of) the data as reported are not correct. In particular, looking at the CO plot in Figure 3, the trends for the two instruments follow one another in a nearly identical manner with overall very similar diurnal profiles. If the NAPS were as sensitive to wind direction as the authors purport, then its corresponding diurnal profile would not manifest the same diurnal trend as the FTIR: The NAPS profile would better reflect the wind direction dependence, yet the NAPS values never trend down as the FTIR values goes up (or vice versa). That is to say, at no time is there an obvious anti-correlation seen in the data. Furthermore, CO is a very well mixed gas and the NAPS instrument (Figure 1) is physically located in the middle of the FTIR optical path. Since the path for the FTIR includes both the highway and a stretch of land greater in length than the highway, the averaged CO mixing ratio obtain by FTIR over the entire path should be similar to the NAPS value, but possible lower due to mixing in cleaner

adjacent air. Inspection of Figure 3 directly points to this – while the magnitudes and offsets differ (significantly!!), both the NAPS and FTIR data maxima and minima track each other very well, having the same diurnal patterns. However, the fact that the CO mixing ratio values (in terms of offset and amplitude) do not agree at a quantitative value suggest that either the data or data analysis is likely incorrect and it is in this regard we criticise the paper.

Specifically, if one observes the CO plot in Figure 3 focusing only on off hours (weekends and during the deep night-time hours), one can draw horizontal lines through the mixing ratio minima for the two methods. While this reviewer is limited to a pencil and paper for the analysis, for such lines "urban minimum baseline" mixing ratio we find the minimum value for the FTIR is $\sim$380 ppb and for the NAPS data the off-hours minimum is $\sim$ 190 ppb, almost exactly a factor of 2.0 difference for this "clean air" baseline. While one could argue that this is due to different dispersion / mixing, this is clearly not the case because: 1) the NAPS point source measurements are $\frac{1}{2}$ the FTIR values (always lower), and the FTIR values should be greater, representing increased dilution of CO, i.e. mixed with cleaner air further from the motorway, and 2) the overall diurnal variations track each other over the entire time interval. We suggest that the difference is likely due to either a systematic negative offset for the CO measurements via NAPS, or (more likely) a systematic large positive offset for the CO measurements with FTIR. It is unclear which of the two instruments is out of calibration, but we suggest the FTIR values are systematically offset and have the incorrect scaling factor, as nowhere in the paper is a calibration procedure reported. There are differences in the offsets for the other gases as well, but CO appears to have the highest. Furthermore, the mixing ratio range for CO from the FTIR is ca. 60% of the NAPS. Again, just by "eyeballing it" the data would appear to show a relative instrument response relation is of the order Y_FTIR = (0.60)*Y_NAPS + 180 ppb. Picking the EDT of 24/7-25/7 (CO from Figure 3), the minima and maxima values for the NAPS are $\sim$ 220 and $\sim$900 ppb, respectively, which is a range of $\sim$ 680 ppb. While the minima and maxima values for the FTIR are $\sim$ 400 and $\sim$780 ppb, respectively, which is a range of $\sim$ 380 ppb or 56% the dynamic

range of the NAPS. Ozone has less of an offset, but the scaling factor is greater. Using the same EDT as above, the dynamic ranges for NAPS and FTIR are 48 and 20 ppb, respectively, which correspond to the FTIR being 42% the range of NAPS. Clearly, there is (are) a systematic flaw(s) present that biases the results by factors of 1.7 (CO) and 2.4 (ozone) Again, since the measurements do not agree, it is unclear what the mixing ratios are for CO and ozone at this certain location at this specific point in time.

Moreover, it is not clear if this scaling / offset artefact is just for CO and ozone or pertains to the analysis of other species as well? Plots of the spectra fit (with residual) are clearly warranted. There may be interferences in the spectra that are causing the offset and the lower dynamic range for CO, yet, there are no spectra in the paper for the readers to observe or evaluate. With no data to observe, we cannot say for certain, but we suggest that the FTIR data are the more suspect of the two instruments. In the paper, there are no explicit data evaluation plots that show what the classic least squares fit looks like relative to the measured spectrum as well as the associated residual plot. It would appear that there was minimal to none of the "hands on" analysis of the FTIR spectra, and as if the results of the Bruker FTIR software (OPS) were used without inspection or vetting. There may thus be miscalculations present that initially went unnoticed. FTIR spectral analysis of gaseous mixtures is not yet a fully automated procedure, but is an interactive process that is subtle and prone to mistakes, interferences, etc. There are several more sophisticated gas analysis programs that may be used to independently confirm or refute the results calculated by OPS, and actual evaluation of the spectra is required.

Furthermore, while authors did correct the raw mixing ratios for temperature and pressure, it appears that the temperature/pressure corrections need to be processed on the reference spectra as well, prior to the fitting. In other words, each reference spectrum (from HITRAN or PNNL) need to be scaled to the correct temperature/pressure then used for quantification. For example, the high resolution reference spectra need to be deresolved to the same resolution of the measured spectra, which in this case is 0.5

cm-1. In the paper, it was not evident if the HITRAN or PNNL reference data were correctly fit to the instrument parameters and instrument lineshape (ILS). It is critical to create the same (instrumental) lineshape and resolution for the fit. Reference spectra may be deresolved using a Gaussian, Lorentzian or Voigt profile, and after deresolving them it is a good idea to check the FWHM to confirm the resolution.

Finally, it appears that one or both instruments were used without any calibration. In order for the scientific community to have confidence in the values obtained, it is important that the instrument(s) undergo some sort of on-site calibration. For the FTIR a wavenumber calibration is necessary and e.g. uses a small gas cells containing NH3 are used for this purpose; other compounds such as H2O, CO or CO2 can also be used. For both IR wavelength and intensity values, the authors are suggested to see e.g. the EPA TO-16 for protocol and procedures. Also to this end, the authors should see the following manuscripts list below, particularly, the many papers of Griffith, Yokelson, Lindenmaier, Burling, Strong, and others where one interactively evaluates and inspects the micro windows associated with specific molecule detection. This is still an involved procedure and there are many papers in the field.

In light of all the calibration issues, we cannot recommend the paper be published as is – while we believe that it affects the ability to quantise all the species, there are too many lingering questions about the reported values, e.g. is the CO mixing ratio 200 ppb, or 400 ppb?

Some (of several) relevant references are listed below.

Bishop, G. A., McLaren, S. E., Stedman, D. H., Pierson, W. R., Zweidinger, R. B., & Ray, W. D. (1996). Method comparisons of vehicle emissions measurements in the Fort McHenry and Tuscarora Mountain Tunnels. Atmospheric Environment, 30(12), 2307-2316.

Burling, I. R., Yokelson, R. J., Griffith, D. W., Johnson, T. J., Veres, P., Roberts, J. M., ... & Hao, W. M. (2010). Laboratory measurements of trace gas emissions from

biomass burning of fuel types from the southeastern and southwestern United States. Atmospheric Chemistry and Physics, 10(22), 11115-11130.

Chaney, Lucian W. "The remote measurement of traffic generated carbon monoxide." Journal of the Air Pollution Control Association 33.3 (1983): 220-222.

Coleman, Marc D., Simon Render, Chris Dimopoulos, Adam Lilley, Rod A. Robinson, Thomas OM Smith, Richard Camm, and Rupert Standring. "Testing equivalency of an alternative method based on portable FTIR to the European Standard Reference Methods for monitoring emissions to air of CO, NOx, SO2, HCl, and H2O." Journal of the Air & Waste Management Association 65, no. 8 (2015): 1011-1019.

Goode, J. G., Yokelson, R. J., Susott, R. A., & Ward, D. E. (1999). Trace gas emissions from laboratory biomass fires measured by open-path Fourier transform infrared spectroscopy: Fires in grass and surface fuels. Journal of Geophysical Research: Atmospheres, 104, 21237.

Griffith D.W.T. and I.M. Jamie, "FTIR Spectrometry in atmospheric and trace gas analysis in Encyclopedia of Analytical Chemistry – Applications, Theory and Instrumentation," R.A. Meyers, Ed. John Wiley and Sons, Ltd. Chichester, (2000).

Griffith, David WT, et al. "FTIR remote sensing of biomass burning emissions of CO2, CO, CH4, CH2O, NO, NO2, NH3, and N2O." Global biomass burning. Atmospheric, climatic, and biospheric implications. 1991.

Griffith, David WT, and Bo Galle. "Flux measurements of NH 3, N 2 O and CO 2 using dual beam FTIR spectroscopy and the flux–gradient technique." Atmospheric Environment 34.7 (2000): 1087-1098.

Grutter, M., Flores, E., Basaldud, R., & Ruiz-Suárez, L. G. (2003). Open-path FTIR spectroscopic studies of the trace gases over Mexico City. ATMOSPHERIC AND OCEANIC OPTICS C/C OF OPTIKA ATMOSFERY I OKEANA, 16(3), 232-236.

Haugen, Molly J., and Gary A. Bishop. "Repeat Fuel Specific Emission Measurements

on Two California Heavy-Duty Truck Fleets." Environmental Science & Technology 51.7 (2017): 4100-4107.

Horrocks, L., Burton, M., Francis, P., & Oppenheimer, C. (1999). Stable gas plume composition measured by OP-FTIR spectroscopy at Masaya Volcano, Nicaragua, 1998-1999. Geophysical Research Letters, 26(23), 3497-3500.

Johnson, T. J.; Profeta, L. T.; Sams, R. L.; Griffith, D. W.; Yokelson, R. L., An infrared spectral database for detection of gases emitted by biomass burning. Vibrational Spectroscopy 2010, 53 (1), 97-102.

Lindenmaier, R., Batchelor, R. L., Strong, K., Fast, H., Goutail, F., Kolonjari, F., ... & Walker, K. A. (2010). An evaluation of infrared microwindows for ozone retrievals using the Eureka Bruker 125HR Fourier transform spectrometer. Journal of Quantitative Spectroscopy and Radiative Transfer, 111(4), 569-585.

Oppenheimer, C., & Kyle, P. R. (2008). Probing the magma plumbing of Erebus volcano, Antarctica, by open-path FTIR spectroscopy of gas emissions. Journal of Volcanology and Geothermal Research, 177(3), 743-754.

Popa, Maria Elena, et al. "Vehicle emissions of greenhouse gases and related tracers from a tunnel study: CO: CO 2, N 2 O: CO 2, CH 4: CO 2, O 2: CO 2 ratios, and the stable isotopes 13 C and 18 O in CO 2 and CO." Atmospheric Chemistry and Physics 14.4 (2014): 2105-2123.

Stremme, W., Grutter, M., Rivera, C., Bezanilla, A., Garcia, A. R., Ortega, I., ... & Hannigan, J. W. (2013). Top-down estimation of carbon monoxide emissions from the Mexico Megacity based on FTIR measurements from ground and space. Atmospheric Chemistry and Physics, 13(3), 1357-1376.

Stedman, Donald H. "Automobile carbon monoxide emission." Environmental Science & Technology 23.2 (1989): 147-149

Yokelson, R. J.; Karl, T.; Artaxo, P.; Blake, D. R.; Christian, T. J.; Griffith, D. W.; Guenther, A.; Hao, W. M., The Tropical Forest and Fire Emissions Experiment: overview and airborne fire emission factor measurements. Atmos. Chem. Phys. 2007, 7 (19), 5175-5196.

Yokelson, R. J.; Christian, T. J.; Karl, T.; Guenther, A., The tropical forest and fire emissions experiment: laboratory fire measurements and synthesis of campaign data. Atmos. Chem. and Phys. 2008, 8 (13), 3509-3527.

Yokelson, R. J., Burling, I. R., Gilman, J. B., Warneke, C., Stockwell, C. E., Gouw, J. D., ... & Kuster, W. C. (2013). Coupling field and laboratory measurements to estimate the emission factors of identified and unidentified trace gases for prescribed fires. Atmospheric Chemistry and Physics, 13(1), 89-116.

Other Suggestions Abstract Pg. 1, sent.21: In previous studies, emission factors have units of g kg-1, here the emission factors have units of g km-1. Please explicitly define the emission factor that you are estimating somewhere in the manuscript. Introduction

Pg. 1 sent 29: There are more pollutants associated with motor vehicles that are not listed here. (Please see Review Article)

Pg. 2 sent 40: change "of" to "in"

Pg. 2 sent 60: change "spectrometry" to "spectroscopy"

Pg. 3 sent 65: remove "however" from the sentence.

Pg. 3 sent 85-86: remove "which were" from the sentence

Pg. 3 sent 87-88: please state what NAPS measures

Pg. 3 sent 89: change "in-situ" to "in situ"

Pg. 3 sent 95: the objectives have already been done (paper by Griffith et al. and Yokelson et al.). Experimental

Pg. 4 sent 104: The Globar temperature between 1200 and 1500°C is its varying state,

however, it will not be in its varying state during the measurements.

Pg. 4 sent 105: This is called a bistatic configuration.

Pg. 4 sent 116: Please add a reference to the end of this sentence. For example: Akagi, S.; Yokelson, R. J.; Burling, I.; Meinardi, S.; Simpson, I.; Blake, D. R.; McMeeking, G.; Sullivan, A.; Lee, T.; Kreidenweis, S., Measurements of reactive trace gases and variable O 3 formation rates in some South Carolina biomass burning plumes. Atmos. Chem. and Phys. 2013, 13 (3), 1141-1165.

Pg. 4 sent 119: Please add references to the previous studies that used absorption features in spectral window for analysis.

Pg. 4 sent 122: Please add reference for the HITRAN database

Pg. 4 sent 123: Please add reference for the PNNL database

Pg. 5 sent 126: How were these values adjusted for temperature/pressure? The reference spectra need to be adjusted to the correct temperature/pressure and used in the fitting process. From this sentence, it appears that the reference spectra were not corrected, but instead the reference spectra were used as is to calculate the mixing ratios, which were then adjusted for temperature/pressure.

Pg. 5 sent131: cite PNNL and HITRAN databases.

Pg. 5 sent 135: This is a huge uncertainty for greenhouse gases. For example, is the $CO_2$ 400? Or 440 ppm?

Pg. 5 sent 146: sensible heat flux. . . what is this?

Pg. 7 sent 174: does this make a difference?

Pg. 7 NAPS measurements: please provide type of analysers used at the NAPS station. Results and Discussion

Pg. 9 sent 225: the measurement are off and so what is the point of the study?

Pg. 9 sentence 244 add references to this sentence.

Pg. 9sentence 247 "generally agree with each other, but with a significant offset..." what does this mean? The language used here is vague and does not tell us anything.

Pg. 10 paragraph1/looking at figure 3: CO from FTIR has an offset of 390 ppb and the NAPS has an offset of 190 ppb?

Pg. 12 sent. 320-321: Please cite the studies that you are referring to here. Throughout the paper whenever referring to studies, please cite them.

Pg. 12 sent. 331: Fig 6 should be Fig 5?

Pg. 13 sent 364: Here you are not comparing the FTIR results (due to water interference) to the model, yet FTIR is mentioned here.

Pg. 13 sent 369: remove "and a" from the sentence.

Pg. 17 sent. 471: Here it states the differences, and the range is large, yet you state that it "within the range of estimates". Cite some of those ranges to support your claim.

Pg. 18 sent. 498: change "are in range" to "in the range"

Pg. 18 sent. 499: change "of" to "the"

Pg. 19 sent. 529: change "comparable" to "compare"

---

## Editor Comment (EC1) · J. Williams (Editor) · 15 Aug 2017

Comments from anonymous reviewer 3.

Long-path measurements of pollutants over a highway in Toronto Yuan You et al.

Summary The paper presents measurements of CO, O3, NH3, HCHO, HCN, CH3OH made by an open path FTIR over highway 401 in Toronto, ON for a 15 day period in July (year was not given). Long path measurements of some compounds were compared to a co-located near road air quality monitoring site and to results from a chemical transport model. Data from a co-located scintillometer was used as input

for a dispersion model (WindTrax) to calculate emission rates from vehicles given the measured pollutant concentrations.

General Comments The authors have useful data to show on roadside levels of NH3 and HCN and potential impact of vehicle emissions as a source of these compounds. For me that is the principle value of this paper. The basic analysis of the data to show the level of agreement between open path (OP) and fixed point measurements is also useful. In general the paper is well written and organized and the figures are clear. I think the analysis of the data is some cases has been stretched to the limit of credibility; in particular the comparison of weekday / weekend effects from such a limited data set. The authors should put their 15 days worth of data into context using the longer record of data from the near road site. A significant part of the paper was calculating CO, NH3, NO, and CH3OH emission factors using the WindTrax dispersion model. This is a free online particle dispersion model but I do not have the expertise to comment on the technical merits of this model and thus this portion of the analysis. A major input to the model is the "background" concentration and I thought choices made for NH3 and CO need better explaining. The authors show that the vehicle emission values they calculate agree reasonably well with ranges reported by others. Emission values reported for methanol are hard to believe as this compound has large sources from other things and is not a major emission that I know of from vehicles. The methanol results merit more discussion and highlights that meteorological variability may induce correlations between compounds that get interpreted as being source driven.

I have identified in the minor comments things that were unclear, some issues I had with the analysis, and a section in the introduction that could be removed. I think if the authors could revise the paper to address some of these issues few issues I have this paper would be it pretty good shape for publication.

Minor Comments P3. It the interest of brevity, the section on page 3 describing how PBL dynamics can impact surface concentration of pollutants is probably unnecessary for the readership of this journal. I found this introductory material unnecessary and I

think it adds to manuscript bloat.

P3. "... first direct comparison of this kind..." It would be good to check the publication of M. Grutter at Centro de Ciencias de la Atmósfera, UNAM, Ciudad Universitaria, Mexio City. He also uses OP FTIR and there had been some big field international air quality field experiments in Mexico City over the last 15 years that would have likely produced opportunities for OP FTIR / fixed point measurement comparisons. I know he has done this for formaldehyde.

P4. Experimental section should list dates of the study period.

P8. It is not clear why the GEM-MACH model results for CO was averaged over 3 hours (1 hour period on each side of the h1-hr period of interest) to get a running average to compare with the 1-hr averages of the data?

P8. WindTrax. The discussion didn't make clear how the concentration at the measurement site was apportioned to the source area (highway lanes) of interest. Wouldn't the back trajectory model need a high resolution emission model to determine what mass of CO measured at the site was from the emission area of interest? This needs to be clarified for the reader who hasn't used WindTrax . Why is this model needed for equation (9) if the denominator is being determined by another model (the bLS model)? I found this section confusing.

Figure 2. I can't tell the difference between the line for z/L and the line for u*.

Fig 4. This is a nice figure but it isn't clear from the text what is actually plotted – the image looks smoothed to color code difference ranges rather than being a collection of individual data points.

P11. Ambient temperature. It is well known that traffic emission of CO can be influenced by temperature but this is primarily due to start emissions when catalytic converters are still cold (< 200 C). Vehicle running emissions of CO are not strongly influenced by ambient temperature. This section has an odd reference "Choi pdf" accessed

from the internet. It would be better to cite an actual EPA report on MOVES temperature parameterization of vehicle emissions. One suggested reference is "MOVES2010 Highway Vehicle Temperature, Humidity, Air Conditioning, and Inspection and Maintenance Adjustments", EPA-420-R-10-027.

P12. Why was a background value of 256 ppbv used for CO – what is the reasoning for this as a "background" values for the airshed or for upwind of the FTIR beam? Do you get the same value for a CO vs NOx regressions? Air entering the urban airshed or crossing the highway will contain NH3 and CO – shouldn't these background values be subtracted from both to reveal increase due to local traffic emissions? This background value is an important number as it is later used in the WindTrax calculations so it deserves better definition.

P14. It is more common in the literature to report CO vs NOx regressions and to discuss CO-to-NOx molar ratios (cf. the papers by D.D. Parrish or Wallace et al Atmos Environ. 2012). A ratio of $\sim 5$ would be expected for running emissions at your site. I think it would be better to show Figure 10 in the traditional way (NOx vs CO) so that your slopes could be compared with the literature and vehicle emission inventory.

P14. The analysis of the weekend / weekday comparison of ozone is perhaps more than what the data can support. There were only 2 weekend periods. Is this really enough data to statistically demonstrate that weekends have different ozone production rates that weekdays? Isn't the production and accumulation of ozone in the airshed also affected by meteorology (irradiance, dispersion)? How were these factors accounted for? You state poor statistics in explaining HCHO patterns. The week day / weekend difference of vehicle emission on ozone production is interesting but you do not have a statistically relevant difference with 15 days of data. This should be recognized in this section. I would recommend you can place the campaign data into context with ozone data from the NAPS site for a multi-year summer period.

P15. If the gas phase mechanism in the GEM-MACH model does not explicitly represent HCHO then it shouldn't be portrayed as HCHO in Figure 6, that is somewhat misleading if one doesn't read the fine print. What other compounds are included with HCHO, methacrolein and methyl vinyl ketone? If this is the case then it I suggest leaving out the model data in Fig 6 for "HCHO".

P15. The HCN section is very brief. Any idea why it is so variable; most data appear below DL of instrument except for 3 days at the end of the campaign. If HCN is from vehicle exhaust why isn't it elevated when CO was elevated? It is hard to tell from the figure, but it doesn't seem to follow CO.

P15. I don't understand the reasoning behind the statement " ...flat on weekends, indicating that a large component of CH3OH may have come from traffic emissions". Methanol doesn't co-vary with CO from examination of the figures. I find it hard to believe all CH3OH in an urban area is due to vehicles. What are other sources of methanol? As far as I know methanol is not included as a compound in vehicle emission inventories by the US EPA but perhaps this is different in Canada? Trees emit methanol. You would probably measure similar levels of methanol outside of the Toronto urban area as a result. Are there other urban sources of methanol that are relevant, solvent use for example?

---

## Author Comment (AC1) · 26 Sep 2017

1. Line 113-line 116: this needs slight elaboration – the stray light problem should be very minor and only from internal reflections (since the light is modulated before being sent across the open-path. This means ambient scattered radiation will not be modulated and thus not detected by the instrument).

Yes, the stray light influence on concentration retrieval is generally rather minor, but since it is possible to correct for this, we did. These sentences in the main text have been revised as follows: "This stray light spectrum accounts for radiation back to the detector from reflections by internal parts inside the spectrometer, i.e. not from the

retroreflector array, and was subtracted from all the measurement spectra before performing further analysis. Stray light affected final mixing ratios by < 3 % in this study."

2. The MALT program (which you say that your analysis is based on) can calculate the reference spectra directly from the HITRAN database using the ambient temperature measured at the same time as the spectra were recorded. Does the Bruker software really not allow this? This may only introduce uncertainties of 10% but it is an unnecessary added uncertainty, since you have accurate temperature measurements available.

This was a mistake in the manuscript; our analysis is based strictly on Bruker software, not MALT. The Bruker software, OPUS_RS, uses a non-linear fitting method. Reference spectra were fitted to the measured spectra using a model to calculate the instrumental line shape (ILS) (Harig et al., 2005) followed by a non-linear curve fitting method to retrieve concentrations of pollutants. In the OPUS_RS software, there is an option of setting up "temperature dependent reference files". These files include either PNNL (5 °C, 25 °C, 50 °C) or HITRAN files at specific temperatures. Then the program takes the current temperature from a sensor (or temperature data file), and interpolates the high-resolution reference spectrum to the current temperature from those temperature-specific reference spectra which were included in "temperature dependent reference files".

Harig, R., Rusch, P., Schäfer, K., Flores-Jardines, E.: "Method for on-site determination of the instrument line shape of mobile remote sensing Fourier transform spectrometers", SPIE 5979, 432-441, 2005.

3. There are other uncertainties inherent to open-path FTIR measurements (like those that come from the HITRAN database and fitting errors). These are not mentioned in the text but should at least be referred to as existing even if a full uncertainty analyses is not given.

Yes, some uncertainties are associated with spectrum fitting errors; the thresholds of the correlation coefficients used in fitting analysis in each pollutant in Table 1 give an

indication of this. There are also uncertainties dependent on environmental conditions. In this study, we have considered ambient temperature and pressure, as well as H2O vapor as an interfering gas, in our retrieval analysis. We have stated that for pollutants such as NO and NO2, water vapor interfered so much that we were not able to get good mixing ratio retrievals. The signal intensity differences related to the distance between spectrometer and retroreflector play a minor role on the detection limits of pollutants studied, since the distance in this study was near the optimal range for this instrument. These detection limits were updated in Table 1 and the text has been revised as follows: "Besides fitting errors and the effect of ambient temperature on the reference spectrum, other environmental conditions may also contribute to uncertainties, such as interference from ambient water vapor."

4. Section 2.2 – at what time resolution are these calculations made? As mentioned in the main text, the LEDs were operated in the continuous mode. H, u* and were calculated at a 1-minute resolution in this study.

To clarify, the main text has been revised: "Sensible heat flux (H), friction velocity (u*) and Obukhov length (L) were calculated from scintillometer measurements at a 1-minute resolution in this study." In addition, to make the manuscript more concise, we decided to move the theory of scintillometer to the Supplementary Material Section 2.

5. Line 212 – Isn't an estimate using WindTrax and CO mole fractions a "top-down" estimate? You then compare it to one based on traffic volumes – isn't this one "bottom-up"?

Yes, that was a typo and has been corrected. WindTrax and CO mixing ratio is a "top-down" estimate. A traffic-volume-based estimate is 'bottom-up'.

6. Line 247: do they generally agree? The level of agreement is not quantified. From looking at the time-series the variations certainly seem to be well captured, but it would be good to give a correlation coefficient for this.

We have done this analysis, and now we include Fig. S2a, CO_NAPS vs. CO_FTIR, in the supplemental material. The r2 is 0.67 when the wind came across the highway to the NAPS trailer, and 0.57 when the wind came from other directions.

7. In fact some basic statistics for the model's skill level would improve the manuscript. The authors have used the "open-air" package and this has some great tools for quick evaluation of a model's performance against observations.

This is a good suggestion. We have added mean bias and root mean square error of comparison between CO measurement and CO model in the supplemental material, Fig. S2b and c.

8. Reading this discussion about model to measurement spatial differences begs the question as to why a comparison of model to open-path FTIR is not shown. This will suffer similar problems but should be much less than the in-situ observations.

In the supplementary material, we have included Fig. S2c on CO_model vs. CO_FTIR with statistical results, and Fig. S4 polar plot of (CO_model – CO_FTIR) vs. wind direction. In the comparison of CO_model and CO_NAPS, the slope when the wind came from the highway and the slope when the wind came from other direction was very different (0.72 vs. 1.20), indicating the strong spatial differences. In the comparison of CO_model and CO_FTIR, the slopes are much closer (1.20 vs. 1.09), indicating the sampling spatial difference is smaller when comparing path-integrated mixing ratio with the volume-averaged chemistry transport model. From the polar plot of CO_model –CO_FTIR vs. wind direction and speed, it also can be seen that positive differences (yellow) occur mainly when the wind is from other directions, and the dependence of (CO_model – CO_FTIR) on the wind direction is not that strong. This discussion is also included as a new paragraph in the main text, Section 3.2.1.

9. Line 297. Are the traffic volumes similar on the weekends? This is surprising and you have not actually stated that clearly before. Can you clarify?

Yes, traffic volumes on the weekend were similar to weekdays except for the morning period (from around 6:00 to 11:00), as shown as the black dashed line in Fig. 5 (bottom). We also wrote "The median CO mixing ratio on weekends was close to that on weekdays, except for the early morning period." And to make it clearer to readers, we have included the actual traffic volume number in this sentence: "On weekends, traffic volume increased more gradually during the morning until plateauing around 11:30 and on average remained high with about 21800 vehicles h-1 until after 22:00."

10. Line 323: is it worth showing the correlation plot at least in the supplementary data?

Agreed. A supplemental figure of linear regression of NH3 and CO mixing ratio from FTIR has been added to the supplemental material (Fig. S5).

11. Line 332: Are you also assuming that traffic is the only source of CO emissions above background?

Yes, we assumed that highway traffic emission at this spatial scale is the only source of CO above background, to estimate traffic-related NH3 emission. We observed a good linear relationship between mixing ratio of NH3 and CO, shown in Fig. S5.

12. Line 438: Why only 3 days? I assume that these are the best steady wind conditions? Whatever the reason for the choice of these days, it should be stated briefly in the text.

We only picked three days because the dispersion model is not easily automated and requires lots of manual labour. We have revised that paragraph and added the following sentences to the text: "July 22 was chosen because the wind direction was steadily from northwest, and a traffic jam occurred for added interest. July 28 and 29 were chosen because they are two of the highest days for temperature and O3 during this project."

13. Line 529: "reasonable" correlations observed. You need to provide some actual

statistics to back this up somewhere in the manuscript.

We meant to point out that wind direction affects the comparison, so we revised this sentence as follows: "Model results and measurement results are not expected to be directly comparable for all wind regimes, and comparisons can be better explained after separating wind directions."

14. Consider changing "mixing ratio" to "mole fraction" throughout, as I believe this is now the preferred terminology.

Thank you. We have seen both terms. This is noted in line 53 in the revised manuscript.

15. Line110: the "fraction" of the path is not actually a fraction but a distance, consider rephrasing.

This was rephrased into "the length of the path that was directly over..."

Please also note the supplement to this comment:
https://www.atmos-chem-phys-discuss.net/acp-2017-328/acp-2017-328-AC1-supplement.pdf

[Figure]

**Supplement:**

**Supplementary material for "Long-path measurements of pollutants and micrometeorology over Highway 401 in Toronto"**

Yuan You[1], Ralf M. Staebler[1*], Samar G. Moussa[1], Yushan Su[2], Tony Munoz[2], Craig Stroud[3], Junhua Zhang[3], and Michael D. Moran[3]

[1]Air Quality Processes Research Section, Environment and Climate Change Canada, Toronto, Ontario, Canada, M3H 5T4.
[2]Ontario Ministry of the Environment and Climate Change, Toronto, Ontario, Canada, M9P 3V6
[3]Air Quality Modelling and Integration Section, Environment and Climate Change Canada, Toronto, Ontario, Canada, M3H 5T4.

*Correspondence to: ralf.staebler@canada.ca

**1. Description of spectrum fitting method**

As mentioned in the main text, high resolution reference spectra were taken from the HITRAN database at 296 K and 101325 Pa when available. For species not available in the HITRAN database, the reference spectra were taken from the PNNL database at 25 $^{\circ}$C. Before measurements, the wavenumber calibration was performed by fitting a function for the instrumental line shape (ILS) (Harig et al., 2005). The ILS function also takes the reference spectra and calculates reference spectra into spectra with the same spectral resolution, 0.5 cm$^{-1}$ in our case. A known gas is selected, water $H_2O$ in our case, and a spectral range is fitted by adjusting ILS parameters.

As described in the main text, "at the beginning of the measurement period, a stray light spectrum was recorded by pointing the spectrometer away from the retroreflector. This stray light spectrum accounts for radiation back to the detector from reflections by internal parts inside the spectrometer, i. e. not from the retroreflector array, and was subtracted from all the measurement spectra before performing further analysis." The software which collects spectra and does spectral fitting analysis is called OPUS_RS. OPUS_RS calculates the spectral fitting by a non-linear curve fitting algorithm. Spectral fitting analysis was mainly done in the process of making reference files for each pollutant. Spectral windows/regions and interference gases for each gases were determined by examining the reference spectrum and picking the absorption features. Spectral windows in previous studies were also examined and adjusted to get a good fit. For CO, $CO_2$, $N_2O$ HCN, HCHO, and $NH_3$, we used spectral regions reported in previous studies (showing in Table 1 in the main text), since the fittings were good. It was also critical to include interference gases relevant in the spectral window under analysis. After many rounds of adjusting windows and interfering gases, an "optimized fit" was determined by examining the measurement spectrum, fitted spectrum, correlation coefficients and residuals. To further improve fittings, baseline calculations were performed under given spectral windows and interfering gases. Three methods are included in the baseline algorithm of the OPUS_RS software: offset, linear baseline correction, and standard baseline algorithm. "Offset" and "linear baseline correction" provide a linear baseline calculation, and they were used when the base of the spectral region was roughly straight. The "standard baseline correction" algorithm uses Gaussian functions to simulate the baseline. It has two parameters: distance and width (both in cm$^{-1}$). "Distance" controls how far apart the centers of multiple Gaussian simulations are and how many Gaussians are within the range of spectral window. "Width" controls the

width of each Gaussian function. "Distance" and "width" parameters were adjusted to optimize fitting. In this study, up to 3 Gaussian functions in the spectral window were used. For each fitting, the OPUS_RS returns a coefficient of correlation. Users have an option of setting a threshold of this coefficient. As noted in Table 1, OPUS_RS reports mixing ratios when the correlation coefficient of fitting is above this threshold. When the correlation is below this threshold, the pollutant is not "identified" and the mixing ratio is reported as zero.

The temperature dependence of the reference spectra was studied for CO, $NH_3$, $CO_2$ and $CH_4$. HITRAN spectra at 278 K, 298 K and 323 K were used in the OPUS_RS. One HITRAN spectrum at one of the three temperatures was loaded into the reference file which contained all the fitting information to produce a reference file for that temperature. The measurement conditions were set to 296 K and 101325 Pa in OPUS_RS, and measurement spectra were loaded by using the reference file at that temperature. The analysis was repeated by using reference files at three temperatures. After fitting the measurement spectra to reference spectra at the three temperatures, the difference in the extracted mixing ratio between 323 K and 278 K divided by the mixing ratio at 278 K was calculated. The maximum difference in retrieved mixing ratio for the 45 °C range was 8.9 % for $NH_3$, 4.2 % for $CH_4$, 8.3 % for CO, and 4.1 % for $CO_2$. This is a conservative estimate for the temperature related uncertainty, since the observed temperature range was much smaller than 45 °C.

Examples of measured spectra, fittings, and residuals are shown in Fig. S1.

**2. Scintillometer theory**

The structure function of the optical refractive index $C_n^2$ is determined from the fluctuations of the light-beam intensities received by the scintillometer receiver. The structure parameter of temperature $C_T^2$ can be derived from $C_n^2$ given the ambient temperature, humidity, pressure, and wavelength (Thiermann and Grassl, 1992). The determination of the sensible heat flux H based on $C_T^2$ needs additional assumptions based on Monin-Obukhov Similarity Theory (MOST).

H is defined as

$$H = -\rho C_p \theta_* u_* \qquad (S1)$$

where $\rho$ is the air density, $C_p$ is the specific heat capacity of air at constant pressure, $\theta_*$ is the temperature scale, and $u_*$ is the friction velocity. $\theta_*$ and $u_*$ are determined by the MOST functions (Wood et al., 2013):

$$\frac{C_T^2 z^{2/3}}{\theta_*^2} = \Psi_H\left(\frac{z}{L}\right) \qquad (S2)$$

$$\frac{\kappa z U(z)}{\ln(\frac{z}{z_0}) u_*} = \Psi_M\left(\frac{z}{L}\right) \qquad (S3)$$

where the Obukhov length L is defined as

$$L = \frac{T}{\kappa g}\frac{u_*^2}{\theta_*} \qquad\qquad (S4)$$

and z is the height above the surface, $z_0$ is the surface roughness length, the von Kármán constant $\kappa$=0.4, g is the gravitational constant, and U(z) is the mean horizontal wind speed. The scaling functions $\Psi_H$ and $\Psi_M$ can be calculated by Eqs. (S5) and (S6) (Thiermann and Grassl, 1992) and Eqs. (S7) (Paulson, 1970) and (S8) (Businger et al., 1971):

$$\Psi_H = 4\beta_1[1 - 7\frac{z}{L} + 75(\frac{z}{L})^2]^{-1/3} \qquad for\ \frac{z}{L} < 0 \qquad (S5)$$

$$\Psi_H = 4\beta_1[1 - 7\frac{z}{L} + 20(\frac{z}{L})^2]^{1/3} \qquad for\ \frac{z}{L} > 0 \qquad (S6)$$

with $\beta_1$=0.86

$$\Psi_M = -2\ln\left(\frac{1+\chi}{2}\right) - \ln\left(\frac{1+\chi^2}{2}\right) + 2\arctan(\chi) - \frac{\pi}{2} \qquad for\ \frac{z}{L} < 0 \qquad (S7)$$

Where $\chi = (1 - 15\frac{z}{L})^{1/4}$

and $\Psi_M = 4.7\frac{z}{L} \qquad for\ \frac{z}{L} > 0 \qquad (S8)$

These calculations were performed with the software provided by Scintec (SRun, version 1.31; see http://www.scintec.com/english/web/scintec/Details/A012000.aspx). The procedure to calculate the sensible heat flux H from measurements is as follows:

An initial $|L|=|L_{ini}|$ = 1000 m is assumed, where the signs of L and H are determined by the simultaneous measurement of the vertical temperature gradient $\Delta T/\Delta z$. Then $\Psi_H$ is calculated from Eqs. (S5) and (S6) using L and path height z. $\theta_*$ can then be calculated with $C_T^2$ using Eq. (S2). Next $\Psi_M$ is calculated from Eqs. (S7) and (S8). Friction velocity $u_*$ is then calculated by Eq. (S3) given the measured wind speed (from the NAPS station) and $z_0$. A new L can then be calculated from Eq. (S4) using the calculated $\theta_*$ and $u_*$. This process is then repeated iteratively until the change in L is smaller than 0.1. Finally, H is calculated from Eq. (S1) using the last calculated values for $\theta_*$ and $u_*$. In the end, H, $u_*$ and L were calculated at a 1-minute resolution in this study.

**3. Multi-year $O_3$ measurements from a regular NAPS station**

It is also recognized that 2 weekends are not statistically enough to show the difference of $O_3$ mixing ratios between weekdays and weekends. The NAPS station which provided measurement results for this manuscript was a recently installed station, so it was there only for that summer. However, there is another nearby NAPS station (Station # 60430) which has been in operation for several years. NAPS station #60430 is located 175 m south of the highway 401 and about 200 m southwest of the NAPS research trailer in Fig.1. Therefore, we have extracted the $O_3$ mixing ratio from Station # 60430 for July and August from 2013 to 2015, to show some multi-year statistics. The diurnal variations of $O_3$ in July to August from 2013 to 2015 are shown in Fig. S7. The median of the diurnal average $[O_3]$ on weekends from 7 am to midnight are higher than median on weekdays, which is consistent with the results from the new research NAPS trailer shown in the main text and Fig.9. There is no resolvable difference in 2013. Therefore, the difference between weekdays and weekends we observed in our 16-day measurement agrees with longer observations in 2014 and 2015, which suggests it is representative.

It is also interesting to compare the 2015 $O_3$ mixing ratio from the NAPS research trailer at the edge of highway with this regular NAPS station. The maximum median $[O_3]$ on weekends (54 ppb) from the trailer at edge was higher than median $[O_3]$ at the regular station (50 ppb). Both stations show differences of 20 – 24 ppb in the median $[O_3]$ between weekdays and weekends.

**References**

Businger, J.A., Wyngaard, J.C., Izumi, Y. and Bradley, E.F.: Flux- profile relationships in the atmospheric surface layer, J. Atmos. Sci., 28, 181-189, 1971.

Harig, R., Rusch, P., Schäfer, K., Flores-Jardines, E.: "Method for on-site determination of the instrument line shape of mobile remote sensing Fourier transform spectrometers", SPIE 5979, 432-441, 2005.

Paulson, C.A.: Mathematical representation of wind speed and temperature profiles in the unstable atmospheric surface layer, J. Appl. Meteorol., 9, 857-861, 1970.

Thiermann, V. and Grassl, H.: The measurement of turbulent surface-layer fluxes by use of bichromatic scintillation, Bound-Lay. Meteorol., 58, 367-389, doi: 10.1007/BF00120238, 1992.

Wood, C.R., Kouznetsov, R.D., Gierens, R., Nordbo, A., Järvi, L., Kallistratova, M.A. and Kukkonen, J.: On the temperature structure parameter and sensible heat flux over Helsinki from sonic anemometry and scintillometry, J. Atmos. Ocean. Tech., 30, 1604-1615, doi: 10.1175/JTECH-D-12-00209.1, 2013.

**Supplemental Figures**

[Figure]

[Figure]

[Figure]

[Figure]

**Figure S1: Examples of measured spectra at 16:01:45 on July 29, and fit modelled spectra made using the optimum spectral windows and OPUS_RS non-linear fitting method. The black curve is the measured spectrum; the red curve is the fit spectrum; and the brown curve is the residual. (a) full measured spectrum; (b) Spectral range for analysis of CO; (c) Spectral range for analysis of $O_3$; and (d) Spectral range for analysis of $NH_3$.**

[Figure]

**Figure S2: Comparison of CO mixing ratios: (a) NAPS and FTIR; (b) GEM-MACH and NAPS (c) GEM-MACH and FTIR. Red dots in (a) and solid circles in (b) and (c) show results when the wind was from the highway towards the NAPS trailer; Blue dots in (a) and open circles in (b) (c) show results when the wind was from other directions; the solid lines in (b) and (c) are the linear regression excluding 3 points when GEM-MACH results were much greater than measurement values; the dashed lines in (b) and (c) are the linear regression for these excluding the 4 most extreme outliers. Mean bias = [GEM_MACH]-[measurements]. RMSE = root mean square error.**

[Figure]

**Figure S3: Gridded NH₃ emissions field used by GEM-MACH. Each grid cell is 2.5 km by 2.5 km in size, and the red dot marks the location of the NAPS trailer. The color scale indicates the magnitude of the emissions of NH₃ (g s⁻¹) in July 2010 from on-road mobile sources (left) and non-mobile sources (right).**

[Figure]

**Figure S4: Polar plots of the difference of CO mixing-ratios between GEM-MACH output and FTIR measurements. Azimuth angle represents wind direction (meteorological convention), and radius indicates wind speed (m s$^{-1}$). The color indicates the CO mixing ratio difference. The center corresponds to the location of the NAPS trailer. The black dashed line shows the orientation of the highway: above this line, the wind came across the highway to the trailer.**

[Figure]

**Figure S5:  Linear relationship between NH₃ and CO mixing ratio from FTIR over the whole measurement period.**

[Figure]

(a)

(b)

(c)

**Figure S6: Correlation between mixing ratio of pollutants and ambient temperature for (a) HCHO; (b) NH₃; (c) CH₃OH**

[Figure]

**Figure S7:** Average weekday and weekend diurnal cycles of mixing ratios of $O_3$ from NAPS station# 60430 in July-August 2013 (top), July-August 2014 (middle), and July-August 2015 (bottom). Solid blue lines are medians and the shaded areas are the interquartile ranges on weekdays; dashed black lines are medians on weekends.

---

## Author Comment (AC2) · 27 Sep 2017

The reviewer's comments are in bold type and our responses are in normal text. We thank this reviewer for his/her careful reading of our manuscript and for the comments. We have added a section to the supplementary material to describe the analysis process in more detail, and to show example spectra and fits.

**1.       The greatest shortcoming of the paper, however, is that there appear to be serious flaws in the data or data analysis, particularly the FTIR spectral analysis. Indeed, it appears that the FTIR data were not analysed at all, but simply taken "as is" from the system, and the IR data appear to have large systematic errors.**

When writing this manuscript, we made the decision not to dwell on the minutiae of the FTIR analysis but to focus on the results. However, if more detail is required, we can certainly provide this. Quite a significant amount of work was done on optimizing the fittings and quantifying concentrations of pollutants. We have now included a section in the supplemental material to describe the FTIR spectral analysis by the OPUS_RS software in more detail. Examples of a measured spectrum and model fit spectra along with residuals for CO, $O_3$, and $NH_3$ are also shown in Fig. S1. To summarize briefly, the software's analysis algorithm is based on an iterative nonlinear fit of the measured IR spectra in compound-specified spectral windows. High-resolution spectra of the target and interfering gases, as well as baseline functions are fitted to those spectral windows in the measured spectra, making use of the known instrument response, the instrument line shape to meet the spectral resolution of the measurement, i.e., 0.5 cm$^{-1}$ in this study. More detailed descriptions and spectrum examples are now given in the supplemental material.

**2.       With regards to novelty of the study, the crux of the paper is to collect FTIR measurements over an urban highway. Such works have been previously reported – see for example work by Bishop et al., Stremme et al., Colman et al., Grutter et al. and Chaney et al, as well as by other analytical methods (for example, the seminal paper by Stedman et al., the tunnel study by Popa et al and the newly released paper by Haugen et al.).That is to say, there exists significant literature regarding traffic emissions, and the present authors need to cite more of these studies, and also need to cite other studies that use open-path FTIR to measure similar compounds from other sources**

We believe our study contains several novel aspects: first, the measurement site is the busiest segment of the busiest highway in North America, as mentioned in the Introduction. Secondly, we conducted measurements for 16 days continuously. Thirdly, what we measured was pollution in the open ambient air over a highway from a real-world, un-manipulated mix of vehicle types. Our study is different from previous tunnel studies, because the air mass is more confined in the tunnel. Mixing processes and solar radiation conditions in the tunnel are different than in the open environment. We also observed and quantified additional pollutants, such as ammonia, formaldehyde, methanol and hydrogen cyanide, for which not much literature exists. These species are not well understood, and their levels can be different at different regions and countries due to different vehicle emission control regulations, fuels, and technologies. Another unique point of this study is that we combined detailed measurement of turbulent mixing (not only wind speed and direction) at the site simultaneous with pollution measurements from the FTIR, to show the role of turbulent mixing on the local air quality. We thank the reviewer for suggesting additional relevant background papers, and they have been cited in the revised manuscript.

**3.      To add a more unique aspect to their study, we suggest that the authors i) emphasize more the aspect of monitoring near such a large motorway (e.g. perhaps there are atmospheric reactions / products only seen due to higher order rate constants requiring very high CO or O3 levels?), and ii) present a more detailed analysis of the instrumental comparison, i.e. the "shoot-out" between the measurements collected from the FTIR and from the NAPS.**

We believe there are many unique aspects to this study. As we have stated in the manuscript and explained in point 2 above, one unique aspect of this study is that we combined direct measurements of mixing ratios of gas-phase pollutants from such a busy segment of highway with detailed micrometeorological and turbulence measurement to show that not only traffic emission rates, but also turbulent mixing conditions in the surface layer play an important role on the accumulation and build-up of air pollutants over this highway. We also observed several interesting pollutants, such as ammonia, formaldehyde, and hydrogen cyanide, which have important implications to atmospheric chemistry and population health, and for which in-situ measurements over a busy highway have not been commonly reported. Another unique aspect is our evaluation of a "top-down" approach to calculate the emission rates of a few primary pollutants with a dispersion model.
The "shoot-out" aspect will be discussed below points 4, 5, 6, 7, and 8.

**4.      We believe (at least some of) the data as reported are not correct. In particular, looking at the CO plot in Figure 3, the trends for the two instruments follow one another in a nearly identical manner with overall very similar diurnal profiles. If the NAPS were as sensitive to wind direction as the authors purport, then its corresponding diurnal profile would not manifest the same diurnal trend as the FTIR: The NAPS profile would better reflect the wind direction dependence, yet the NAPS values never trend down as the FTIR values goes up (or vice versa). That is to say, at no time is there an obvious anti-correlation seen in the data. Furthermore, CO is a very well mixed gas and the NAPS instrument (Figure 1) is physically located in the middle of the FTIR optical path. Since the path for the FTIR includes both the highway and a stretch of land greater in length than the highway, the averaged CO mixing ratio obtain by FTIR over the entire path should be similar to the NAPS value, but possible lower due to mixing in cleaner adjacent air. Inspection of Figure 3 directly points to this – while the magnitudes and offsets differ (significantly!!), both the NAPS and FTIR data maxima and minima track each other very well, having the same diurnal patterns. However, the fact that the CO mixing ratio values (in terms of offset and amplitude) do not agree at a quantitative value suggest that either the data or data analysis is likely incorrect and it is in this regard we criticise the paper.**

The long-path FTIR provides a path-integrated concentration, whereas the NAPS trailer measured concentration at a point located at the south edge of the highway. Therefore, concentration measured by the NAPS trailer was more dependent on the wind direction than concentration obtained from the FTIR. But we must keep in mind that even a point measurement has a footprint. The footprint covers more highway when the wind comes from the north, and covers much less highway (but more parking lots, buildings, a park and golf club), when the wind was from the south. The polar plot of the (CO_FTIR – CO_NAPS) in Fig. 4a already clearly shows the dependence of the difference between the two measurements on the wind direction. We have stated that when the wind direction was from highway to NAPS trailer, the difference was smaller, and when the wind was from other directions, the difference

was greater. When the wind was from the south, the NAPS trailer measured less footprint covering the highway, but still measured some traffic pollutants. Therefore, the NAPS measurements of CO (primary pollutant) still demonstrated a similar diurnal cycle on weekdays as the FTIR measurements. There is no reason to think that CO_NAPS and CO_FTIR should be exactly anti-correlated, since both instruments sample overlapping plumes.

The FTIR measures a path-integrated concentration, and its footprint was not completely independent of wind direction. The dependence is weaker compared to the NAPS trailer. As the reviewer has noticed, the path of our set up included "a stretch of land", and this stretch of land was downwind when the wind came from north. Therefore, the downwind length of the FTIR path was longer when the wind came from north. To illustrate the different footprints the FTIR picked up with different wind directions, a plot based on the geometry of the path is shown below:

[Figure]

**Figure: Dependence of the footprint of the FTIR path measurements on wind direction.**

Regardless of wind direction, the FTIR path always picked up some pollution emitted from the highway, and this segment of highway always has traffic (daily minimum is around 3500 vehicle per hour at around 2 to 4 a.m.). So the FTIR measurement never actually sees mixing ratios at the level of urban background air. The NAPS trailer observed a lower pollution level than the FTIR when the wind came from the south because the footprint does not include the highway. This explains the offset between minimum levels of

CO_FTIR and CO_NAPS. Again, CO_NAPS and CO_FTIR were not expected to be anti-correlated at any time or wind direction.

In the supplementary material Fig. S1, we have included examples of a spectrum, and spectral fitting of CO, $O_3$, and $NH_3$, at 16:01:45 on July 29, when the wind came from south and a significant offset was observed. These plots of fittings do not show any significant residuals or problems.

We also have checked that NAPS measurement of CO had zero calibrations four times each day. The average of the zero readings was 12.6 ppb with a standard deviation of 7.4 ppb. So the NAPS measurement of CO did not have significant offset and we estimate the uncertainty of CO_NAPS to be 22 ppb (three times of the standard deviation during zero calibration).

[Figure]

**Figure: Time series of CO_FTIR, CO_NAPS, wind speed and wind direction.**

**5.       Specifically, if one observes the CO plot in Figure 3 focusing only on off hours (weekends and during the deep night-time hours), one can draw horizontal lines through the mixing ratio minima for the two methods. While this reviewer is limited to a pencil and paper for the analysis, for such lines "urban minimum baseline" mixing ratio we find the minimum value for the FTIR is ~380 ppb and for the NAPS data the off-hours minimum ~190 ppb, almost exactly a factor of 2.0 difference for this "clean air" baseline.**

As stated above, there really was no urban minimum baseline for the FTIR CO mixing ratio, since it always picked up pollution from the highway, even at night, with a minimum of 3500 vehicles passing by per hour. Therefore, over this extremely busy highway, the concept of "off hours" does not apply. For NAPS, if the wind came from the south, it could get some significantly lower levels than the FTIR.

**6.    While one could argue that this is due to different dispersion / mixing, this is clearly not the case because: 1) the NAPS point source measurements are 1/2 the FTIR values (always lower), and the FTIR values should be greater, representing increased dilution of CO, i.e. mixed with cleaner air further from the motorway, and**

This is not quite correct; the NAPS measurements were not always lower than FTIR. For example, 7/24 around 01:00, 7/26 around 03:00, and 7/27 around 0:30, NAPS and FTIR CO were very close. These periods may correspond to the "off hours" mentioned by the reviewer. However, we think there are no "off hours", since this segment of highway always has traffic (daily minimum is around 3500 vehicle per hour at around 2 to 4 am). NAPS and FTIR CO were very close during these periods mainly because the wind came from north during these periods. It has been shown in the polar plot Fig. 4a.

**7.    2) the overall diurnal variations track each other over the entire time interval. We suggest that the difference is likely due to either a systematic negative offset for the CO measurements via NAPS, or (more likely) a systematic large positive offset for the CO measurements with FTIR. It is unclear which of the two instruments is out of calibration, but we suggest the FTIR values are systematically offset and have the incorrect scaling factor, as nowhere in the paper is a calibration procedure reported. There are differences in the offsets for the other gases as well, but CO appears to have the highest. Furthermore, the mixing ratio range for CO from the FTIR is ca. 60% of the NAPS. Again, just by "eyeballing it" the data would appear to show a relative instrument response relation is of the order Y_FTIR = (0.60)\*Y_NAPS + 180 ppb. Picking the EDT of 24/7-25/7 (CO from Figure 3), the minima and maxima values for the NAPS are ~ 220 and ~900 ppb, respectively, which is a range of ~ 680 ppb. While the minima and maxima values for the FTIR are~ 400 and ~780 ppb, respectively, which is a range of ~ 380 ppb or 56% the dynamic range of the NAPS.**

We agree with the reviewer that the FTIR CO "offset" seems suspicious; this is something that bothered us from the beginning. Because of this, we have expended significant effort to ensure that there are no biases introduced by our spectral analysis. An example fit is shown in Fig. S1. Interference by $H_2O$ was investigated and found to be insignificant; and different spectral windows suitable for CO were investigated ($2068 – 2198$ cm$^{-1}$, with correlation coefficient threshold = 0.7, the difference is 14 to 26 ppb (25 to 75 percentile)), but all led to the same result. After eliminating the possibility of instrumental biases, the most logical explanation remaining is that the higher offset in the FTIR CO is due to the persistent traffic throughout the night, which always affects at least part of the FTIR optical path.

The issue of the slope of 0.6 (rather than 1.0) is again explained by path-integration vs. point measurement. Due to the proximity of NAPS to the highway, larger maxima in CO_NAPS are to be expected, while the CO_FTIR is always an average that includes diluted air.

We have revised the text in the manuscript that describes how we came to the conclusion that this CO offset between FTIR and NAPS appears to be real:

"A major contributing reason for the differences of CO mixing ratios between the FTIR and the NAPS is that the FTIR and the NAPS were not sampling the exact same air, i.e., the measurements represented different footprints. The FTIR measured the air along the path across and above Highway 401, which always included some pollutants emitted from traffic. In contrast, NAPS numbers represented point measurements beside the south edge of the highway. Therefore, CO mixing ratios measured by the NAPS trailer were more dependent on the wind direction than mixing ratios obtained from the FTIR. When the wind was from the south and towards the highway, the NAPS trailer was mostly blind to the highway; when the wind was from the north, it was immediately downwind it. Therefore, CO mixing ratios from the NAPS are expected to be lower than mixing ratios obtained from the FTIR when the wind is from the south and towards the highway.

The path-integrating approach of the FTIR also has a dilution effect since a significant fraction of the path is not above the source (i.e. the highway). Therefore, the CO mixing ratios obtained from the FTIR should be less than CO mixing ratios from NAPS during winds from the highway towards the NAPS trailer. The polar plot in Fig. 4a clearly shows the dependence of the CO mixing-ratio difference between the FTIR and the NAPS on wind direction. When the wind came from the north over the highway towards the NAPS trailer (above the dashed line), CO mixing ratios from the FTIR were close to or lower than mixing ratios from the NAPS. When the wind was from the south and towards the trailer (below the dashed line), the CO mixing ratios from FTIR were higher than CO mixing ratios from NAPS."

**8.       Ozone has less of an offset, but the scaling factor is greater. Using the same EDT as above, the dynamic ranges for NAPS and FTIR are 48 and 20 ppb, respectively, which correspond to the FTIR being 42% the range of NAPS. Clearly, there is (are) a systematic flaw(s) present that biases the results by factors of 1.7 (CO) and 2.4 (ozone) Again, since the measurements do not agree, it is unclear what the mixing ratios are for CO and ozone at this certain location at this specific point in time.**

If one calls the difference between point measurements and path-integrated measurements a systematic flaw, then we agree. They cannot be directly compared in the sense of a cross-calibration; differences are due to the fundamental difference in the measurement itself, and as previously discussed, the observed differences can be explained by the differing footprints, point measurement right next to the highway vs. path-integrated/diluted measurement incorporating 1/3 highway and 2/3 parking lot, which, depending on wind direction, will contain highway emissions or not.

The offset of $O_3$ is mainly due to the different footprint. As already shown in polar plot Figure 7 (a), the difference is small when the wind came from the north, and the main offset occurred when wind came from the south. A time series plot of $O_3$ with wind direction and speed is also shown below. Both polar plot and time series show that the big "offset" occurred when the wind came from around 120 to 220 degrees. The two measurements were close during July 21 to 24 when the wind was steadily from north. We also admit, as indicated in the updated Table 1, that we have less confidence in the accuracy of the FTIR measurements of $O_3$.

[Figure]

**Figure: Time series of O₃_FTIR, O₃_NAPS, wind speed and wind direction.**

**9.      In the paper, there are no explicit data evaluation plots that show what the classic least squares fit looks like relative to the measured spectrum as well as the associated residual plot. It would appear that there was minimal to none of the "hands on" analysis of the FTIR spectra, and as if the results of the Bruker FTIR software (OPS) were used without inspection or vetting. There may thus be miscalculations present that initially went unnoticed. FTIR spectral analysis of gaseous mixtures is not yet a fully automated procedure, but is an interactive process that is subtle and prone to mistakes, interferences, etc. There are several more sophisticated gas analysis programs that may be used to independently confirm or refute the results calculated by OPS, and actual evaluation of the spectra is required.**

As already mentioned in our response to comment point #1, when we wrote the manuscript we made the decision not to get into all the details of the FTIR analysis but to focus on the results. To address the reviewer's concerns, we have included a section in the supplemental material to describe the FTIR spectral analysis by the OPS software. Supplemental text and Fig. S1 have been included to show details of fitting analysis and example spectrum, fitting results, and residual plots. The selected example plot is a spectrum taken at 16:01:45 on July 29, when the reviewer observed a big "systematic offset" of CO and O₃. The wind was from the south. Fig. S1 shows reasonable fits and small residuals of CO, O₃, and NH₃.

**10.     Furthermore, while authors did correct the raw mixing ratios for temperature and pressure, it appears that the temperature/pressure corrections need to be processed on the reference spectra as well, prior to the fitting. In other words, each reference spectrum (from HITRAN or PNNL) need to be scaled to the correct temperature/pressure then used for quantification. For example, the high resolution reference spectra need to be deresolved to the same resolution of the measured spectra, which in this case is 0.5 cm-1. In the paper, it was not evident if the HITRAN or PNNL reference data were correctly fit to the instrument parameters and instrument lineshape (ILS). It is critical to create the same (instrumental) lineshape and resolution for the fit. Reference spectra may be deresolved using a Gaussian, Lorentzian or Voigt profile, and after deresolving them it is a good idea to check the FWHM to confirm the resolution.**

Actually the ILS fit was indeed performed and is included in the quantitative evaluation. We have now included ILS fit information in the supplemental material. Each reference spectrum (from HITRAN or PNNL) was converted to a spectrum with the same resolution (0.5 cm$^{-1}$ in this study) by the ILS model in OPS software. We did not calculate new reference spectra based on the actual temperature before fitting, but we provided conservative estimations (8.9 % for $NH_3$, 4.2 % for $CH_4$, 8.3 % for CO, and 4.1 % for $CO_2$) of temperature related uncertainty of the retrieved mixing ratios in the range of 5 to 50 $^{o}$C, which is much wider than the range of difference between ambient temperature and 296 K during our study. We also corrected the raw mixing ratio from FTIR due to the change of air density with change of ambient temperature and pressure.

**11.     Finally, it appears that one or both instruments were used without any calibration. In order for the scientific community to have confidence in the values obtained, it is important that the instrument(s) undergo some sort of on-site calibration. For the FTIR a wavenumber calibration is necessary and e.g. uses a small gas cells containing NH3 are used for this purpose; other compounds such as H2O, CO or CO2 can also be used.**

A wavenumber/line position calibration was performed by using the spectrum fitting analysis of $H_2O$ from the ambient air. In the Bruker software, OPUS_RS, this process is included in the ILS function, as described in the supplementary material. A gas-cell calibration is of course desirable but is not nearly as straightforward as it sounds, since specialized cell windows are required, making this an expensive proposition.

**12.     Abstract Pg. 1, sent.21: In previous studies, emission factors have units of g kg-1, here the emission factors have units of g km-1. Please explicitly define the emission factor that you are estimating somewhere in the manuscript.**

One sentence has been added to the text in Section 3.7 indicating the units of emission factor reported in the references were gram mile$^{-1}$ and gram km$^{-1}$, and they were converted into gram km$^{-1}$. The references in Table 2 also reported emission factors in grams per distance, and they were all converted into grams km$^{-1}$ to compare to our results. In the last third paragraph in Section 3.7, one sentence was added to explain again that the emission factors results shown in Table 2 were calculated from the emission rate estimates obtained from WindTrax. These emission factors were calculated to compare with previous reported values.

**13. Pg. 1 sent 29: There are more pollutants associated with motor vehicles that are not listed here. (Please see Review Article)**

Yes, thank you. We have written it out more clearly (such as semi- and low-volatile organic compounds, aromatics and PAHs) and cited more references.

**14. Pg. 2 sent 40: change "of" to "in"**

It has been corrected.

**15. Pg. 2 sent 60: change "spectrometry" to "spectroscopy"**

It has been corrected.

**16. Pg. 3 sent 65: remove "however" from the sentence.**

We keep "however" here, since we are contrasting our study to those in the previous paragraph.

**17. Pg. 3 sent 85-86: remove "which were" from the sentence**

It has been removed.

**18. Pg. 3 sent 87-88: please state what NAPS measures**

This sentence has been revised in more specific: "to conduct an in-depth comparison of a range of pollutants ($CO$, $O_3$ and $NO_x$) measured by both the path-integrating FTIR instrument and the in -situ station."

**19. Pg. 3 sent 89: change "in-situ" to "in situ"**

It has been corrected.

**20. Pg. 3 sent 95: the objectives have already been done (paper by Griffith et al. and Yokelson et al.).**

The description of objective point 1 has been revised:  "1) to evaluate the capabilities of the long-path FTIR spectroscopy for quantifying the mixing ratios of gaseous pollutants in a heavily polluted open urban traffic environment for a length of time sufficient to cover a range of environmental conditions (16 days)"

**Experimental**

**21. Pg. 4 sent 104: The Globar temperature between 1200 and 1500C is its varying state, however, it will not be in its varying state during the measurements.**

The exact temperature is not well known, so we decided to leave it out.

**22. Pg. 4 sent 105: This is called a bistatic configuration.**

Thank you. It has been noted in the revised manuscript.

**23.     Pg. 4 sent 116: Please add a reference to the end of this sentence. For example: Akagi, S.; Yokelson, R. J.; Burling, I.; Meinardi, S.; Simpson, I.; Blake, D. R.; McMeeking, G.; Sullivan, A.; Lee, T.; Kreidenweis, S., Measurements of reactive trace gases and variable O 3 formation rates in some South Carolina biomass burning plumes. Atmos. Chem. and Phys. 2013, 13 (3), 1141-1165.**

We don't think this reference is supportive to that sentence, so we added this reference to the biomass burning sentence in line 65 in the revised manuscript.

**24.     Pg. 4 sent 119: Please add references to the previous studies that used absorption features in spectral window for analysis.**

Yes, I have added a column for references to Table 1.

**25.     Pg. 4 sent 122: Please add reference for the HITRAN database**

I had the reference at line 56, and I added the reference here again this time.

**26.     Pg. 4 sent 123: Please add reference for the PNNL database**

I had a reference at line 57; I added this reference here again this time. I also included Johnson et al. (2010).

**27.     Pg. 5 sent 126: How were these values adjusted for temperature/pressure? The reference spectra need to be adjusted to the correct temperature/pressure and used in the fitting process. From this sentence, it appears that the reference spectra were not corrected, but instead the reference spectra were used as is to calculate the mixing ratios, which were then adjusted for temperature/pressure.**

The reference spectra were not corrected by ambient temperature in the analysis shown in this manuscript. Mixing ratios were calculated by reference spectrum from two databases at 296 K and 298 K, and then were corrected by the density of air depending on ambient temperature and pressure. The effect of temperature on the reference spectrum is small for the range of ambient temperature in this study. We estimated the uncertainty due to the difference between the ambient temperature and reference spectrum in a conservative way, as described in section 2.1 and supplementary material.

**28.     Pg. 5 sent131: cite PNNL and HITRAN databases.**

They are cited here now. They were already cited in line 56 and 57 in page 2.

**29.     Pg. 5 sent 135: This is a huge uncertainty for greenhouse gases. For example, is the CO2 400? Or 440 ppm?**

As we have stated in the text, the maximum uncertainty of 10 % is a conservative estimation because we used a much wider temperature range than the actual ambient temperature range. In addition, we did not report any $CO_2$ mixing ratio in the manuscript.

**30.     Pg. 5 sent 146: sensible heat flux. . . what is this?**

Sensible heat flux is the turbulent flux of "heat" across a warm-to-cold air gradient. It is a component of the atmosphere/surface energy budget (net radiation = sensible + latent heat + soil heat flux). It is typically measured using eddy covariance, but in this case, we are calculating it using similarity theory, based on the scintillometer data. (Stull, 2003)

Stull, R.B.: An Introduction To Boundary Layer Meteorology, Kluwer Academic Publishers, Netherland, 670 pp., 2003.

**31.     Pg. 7 sent 174: does this make a difference?**

The location of the weather station was written out here to describe the source of solar radiation data. 9 km is close enough at the relevant time scales.

**32.     Pg. 7 NAPS measurements: please provide type of analysers used at the NAPS station.**

Information on the NAPS analyzers has been added to Section 2.3 as follows: "The CO analyzer [Model 48iTrace level-Enhanced, Thermo Fisher Scientific, USA], operates based on infrared absorption and gas filter correlation; the NOx analyzer [Model 42i Trace Level, Thermo Fisher Scientific, USA] on chemiluminescence; the  $O_3$ analyzer [Model 49i, Thermo Fisher Scientific, USA]  on UV absorption; the $SO_2$ analyzer, [Model 43i, Thermo Fisher Scientific, USA] on UV fluorescence.; and the $PM_{2.5}$ analyzer [Model SHARP 5030, Thermo Fisher Scientific, USA] on light scattering and beta attenuation. Meteorological parameters including air temperature, pressure, relative humidity, and wind speed and direction, were monitored using a WXT520 weather station [Vaisala, Finland]."

**Results and Discussion**

**33.     Pg. 9 sent 225: the measurements are off and so what is the point of the study?**

As we have explained in points 3, 4, and 5, the offset between two measurements can be explained. They agreed well when wind came from north (highway). The main points of this study have been explained in our responses to comments 2 and 3. Text has been added in lines 89-95 on page 3 in the revised manuscript.

**34.     Pg. 9 sentence 244 add references to this sentence.**

Yes, these references have been included now**.**

**35.      Pg. 9sentence 247 "generally agree with each other, but with a significant offset. . ." what does this mean? The language used here is vague and does not tell us anything.**

I have revised this sentence in the text to be more specific: "As shown in Fig. 3, many mixing ratios peaks of CO from the FTIR and the NAPS matched well, and mixing ratios generally correlated with each other

(Fig. S2 a), but with a significant offset and amplitude difference when the wind came from the south (more detailed comparison in the next paragraph)."

**36.    Pg. 10 paragraph1/looking at figure 3: CO from FTIR has an offset of 390 ppb and the NAPS has an offset of 190 ppb?**

Please see our response above in points 4, 5, 6 and 7.

**37.    Pg. 12 sent. 320-321: Please cite the studies that you are referring to here. Throughout the paper whenever referring to studies, please cite them.**

Yes, now I have cited there again. I meant the two references/studies discussed in line 321-330.

**38.    Pg. 12 sent. 331: Fig 6 should be Fig 5?**

Yes, thanks. We have corrected it into Fig. 5 in the revised manuscript.

**39.    Pg. 13 sent 364: Here you are not comparing the FTIR results (due to water interference) to the model, yet FTIR is mentioned here.**

Yes, the reference to the FTIR path has been removed.

**40.    Pg. 13 sent 369: remove "and a" from the sentence.**

Sorry, a typeset mistake. Should have been "significant decrease in the middle of the day and a secondary peak…" It has been revised: " … reaching a peak over 100 ppb from 6:00 to 8:00 followed by significant decrease in the middle of the day and a secondary peak between 20:00 and 23:00."

**41.    Pg. 17 sent. 471: Here it states the differences, and the range is large, yet you state that it "within the range of estimates". Cite some of those ranges to support your claim.**

The ranges have been cited there.

**42.    Pg. 18 sent. 498: change "are in range" to "in the range"**

It has been corrected.

**43.    Pg. 18 sent. 499: change "of" to "the"**

It has been corrected.

**44.    Pg. 19 sent. 529: change "comparable" to "compare"**

This sentence has been revised to: " …model results and measurement results are not expected to directly compare for all wind regimes, and comparisons can be better explained after separating wind directions."

---

## Author Comment (AC3) · 27 Sep 2017

The reviewer's comments are in bold type and our responses are in normal text. We thank the reviewer for his/her careful reading of our manuscript and for the comments.

**General comments:**

**1.      I think the analysis of the data is some cases has been stretched to the limit of credibility; in particular the comparison of weekday / weekend effects from such a limited data set. The authors should put their 15 days worth of data into context using the longer record of data from the near road site.**

The authors recognize that 4 days on weekends are not statistically robust enough to show the difference between weekdays and weekends, especially for $O_3$. Therefore, we extracted the summer $O_3$ data in 2013, 2014, and 2015 from a regular NAPS station near the highway. Please see the details in point 12 below, the revised manuscript, and the supplemental material.

**Minor comments:**

**2.      P3. It the interest of brevity, the section on page 3 describing how PBL dynamics can impact surface concentration of pollutants is probably unnecessary for the readership of this journal. I found this introductory material unnecessary and I think it adds to manuscript bloat.**

We believe this section introduces an essential aspect of our study, and would like to keep it in. We have shortened it by removing 3 sentences. We have also moved most of section 2.2 to the supplementary material.

**3.      P3. "... first direct comparison of this kind..." It would be good to check the publication of M. Grutter at Centro de Ciencias de la Atmósfera, UNAM, Ciudad Universitaria, Mexio City. He also uses OP FTIR and there had been some big field international air quality field experiments in Mexico City over the last 15 years that would have likely produced opportunities for OP FTIR / fixed point measurement comparisons. I know he has done this for formaldehyde.**

We thank the reviewer for the references. Yes, Grutter et al. (2005) did a comparison of op-FTIR with a point measurement at the same site for formaldehyde mixing ratio in downtown Mexico City in 2003. They also did a comparison of op-FTIR measurement of CO with a point measurement of CO at a different site.

We have included Grutter et al. (2005) in the main text. At the end of 5[th] paragraph in the Introduction, "Grutter et al. (2005) measured the formaldehyde mixing ratio by op-FTIR in downtown Mexico City in 2003 and compared it with a point measurement at the same site." And in the second last paragraph of Introduction, "To our knowledge, (Grutter et al., 2005) presents the first of very few direct comparisons of this kind to be published."

**4.      P4. Experimental section should list dates of the study period.**

The dates of the study have been added to the first paragraph in the Experimental section.

**5.     P8. It is not clear why the GEM-MACH model results for CO was averaged over 3 hours (1 hour period on each side of the h1-hr period of interest) to get a running average to compare with the 1-hr averages of the data?**

This is a misunderstanding. GEM-MACH model outputs were not averaged over 3 hours. This version of GEM-MACH produced concentration of pollutants every hour hh:00 (a snapshot), and we wanted an averaged concentration over 60 minutes, so we took two consecutive outputs from GEM-MACH, averaged these two outputs, and used this as the average concentration for the 60 minutes between these snapshots. This has been described in the second paragraph of Section 2.4.

**6.     P8. WindTrax. The discussion didn't make clear how the concentration at the measurement site was apportioned to the source area (highway lanes) of interest. Wouldn't the back trajectory model need a high resolution emission model to determine what mass of CO measured at the site was from the emission area of interest? This needs to be clarified for the reader who hasn't used WindTrax. Why is this model needed for equation (9) if the denominator is being determined by another model (the bLS model)? I found this section confusing.**

WindTrax is a software tool which incorporates a backward Lagrangian Stochastic (bLS) model (Flesch et al., 1995) with a graphical interface. In this interface, we defined the paved surface of Highway 401 as the only source of CO and other primary pollutants listed in Table 2. The bLS model then releases a large number of virtual particles from the point of measurement and calculates individual trajectories backward in time, based on the input meteorological conditions (wind direction, $u_*$, L, temperature). The fraction of trajectories that originate from the defined source area is then determined, and used to calculate the simulated relationship between source strength and concentration, as given in the denominator of equation (1). (Please note that equation (9) in the previous version of the manuscript in now equation (1) in the revised manuscript, since we decided to put the theory and equations for the scintillometer in the supplementary material.) This equation can then be used to calculate the actual source strength, given the observed and background concentrations.

To make it clear to readers, Section 2.5 has been revised. "We used a backward Lagrangian Stochastic (bLS) model (WindTrax, http://www.thunderbeachscientific.com ; Flesch et al., 1995) which calculates the emission rate $Q$ though formula (1) where C is the concentration of a pollutant at the measurement site, $C_b$ is the background concentration, and $(C/Q)_{sim}$ is the simulated ratio of concentration at the site to the emission rate upwind."

**7.     Figure 2. I can't tell the difference between the line for z/L and the line for $u_*$.**

The color of the line for $u_*$ has been changed to red, to distinguish it from z/L.

**8.     Fig 4. This is a nice figure but it isn't clear from the text what is actually plotted – the image looks smoothed to color code difference ranges rather than being a collection of individual data points.**

Figure 4 shows the difference of CO concentrations (as the color code) between FTIR measurements and NAPS measurements, as well as GEM-MACH model results and NAPS measurements. Yes, it is correct that the color has been smoothed. The angle is the wind direction, and the radius is the wind speed. We used this figure to show that the difference of concentration from these two sources depends on the wind direction (i. e. across the highway or not). Since the color smoothing may be confusing to other readers as well, we have changed these figures back to show individual data points.

**9.       P11. Ambient temperature. It is well known that traffic emission of CO can be influenced by temperature but this is primarily due to start emissions when catalytic converters are still cold (< 200 C). Vehicle running emissions of CO are not strongly influenced by ambient temperature. This section has an odd reference "Choi pdf" accessed from the internet. It would be better to cite an actual EPA report on MOVES temperature parameterization of vehicle emissions. One suggested reference is "MOVES2010 Highway Vehicle Temperature, Humidity, Air Conditioning, and Inspection and Maintenance Adjustments", EPA-420-R-10-027.**

Thank you! This EPA report has been added to the references of the revised manuscript.

**10.       P12. Why was a background value of 256 ppbv used for CO – what is the reasoning for this as a "background" values for the airshed or for upwind of the FTIR beam? Do you get the same value for a CO vs NOx regressions? Air entering the urban airshed or crossing the highway will contain NH3 and CO – shouldn't these background values be subtracted from both to reveal increase due to local traffic emissions? This background value is an important number as it is later used in the WindTrax calculations so it deserves better definition.**

In the revised supplemental material, we have added the plot of $[NH_3]$ vs. $[CO]$ from FTIR results (Fig. S5). From the linear relationship analysis, 265 ppb is the intercept of $[CO]$ ($[CO]$ = 256 ppb when $[NH_3]$ = 0 ppb). Our reasoning was that we assumed that traffic emissions were the only source of CO above background at this spatial scale, and all the $NH_3$ from traffic emissions were associated with CO. The slope of the linear regression between $[NH_3]$ and $[CO]$ has also been reported by Livingston et al. (2009) and Perrino et al. (2002). We have compared our slope with those slopes, as discussed in Section 3.3 of our manuscript.

From our FTIR observations, $[NH_3]$ reached 0 ppb sometimes (see time series of $[NH_3]$ in Fig. 6), so we did not consider $NH_3$ background from other sources in this analysis. We used a linear relationship between ($[CO]$-$[CO\_background]$) and $[NH_3]$, to estimate $NH_3$ from traffic emissions, which is shown in Fig. 5. We stated " Overall, there is no indication of a background offset of $NH_3$, and most measured $NH_3$ at this site can be accounted for by traffic emissions " in the manuscript.

As the reviewer suggested, we did another regression analysis: $[CO]$ vs. $[NO_x]$ from NAPS measurement. The intercept of CO on weekdays and weekends are 196 and 186 ppb. This is discussed in point 11 below.

**11.       P14. It is more common in the literature to report CO vs NOx regressions and to discuss CO-to-NOx molar ratios (cf. the papers by D.D. Parrish or Wallace et al Atmos Environ. 2012). A**

**ratio of ~5 would be expected for running emissions at your site. I think it would be better to show Figure 10 in the traditional way (NOx vs CO) so that your slopes could be compared with the literature and vehicle emission inventory.**

As mentioned in Section 3.4, we did not retrieve reliable mixing ratios of NO and $NO_2$ from the FTIR, so we could not do the same regression analysis of [CO] vs. $[NO_x]$ for FTIR measurements. However, we did a regression analysis of $[NO_x]$ vs. [CO] from NAPS measurements as shown in Fig. 10. These slopes have been compared to previous values in Kim et al. (2016) in Section 3.4 in the discussion manuscript. There is another recent publication, Hassler et al. (2016) (D. D. Parrish is one of the authors) showing a long-term observation of $[NO_x]$ and [CO]. In that paper, they showed $[NO_x]$ / [CO] from LA basin observations as well as from near road monitoring stations from Paris and London. Our slopes of 0.1- 0.2 agree well with observation data from LA Basin after 2010.

Figure 10 is in the orientation of $[NO_x]$ vs. [CO] already. Perhaps the reviewer meant a plot of [CO] vs. $[NO_x]$? We did another regression analysis with NAPS measurement [CO] vs. $[NO_x]$. The slopes are 3.14 and 7.75 for weekdays and weekends, respectively, i.e., bracketing the expected slope of 5. In Parrish et al. (2002), the slopes are in the range of 6.3 to 18.9, given data from 1987 to 1999. In Wallace et al. (2012), the slope is 4.2 from morning rush hours in Meridian, Idaho, USA in 2008- 2009.

The ratio of $[NO_x]$ / [CO] depends on the mix of vehicle types, fuels and engines, driving cycles, mileage, and emission control techniques. Figure 10 also shows the difference of $[NO_x]$ / [CO] between weekdays and weekend, due to decreased numbers of heavy-duty diesel vehicles on the weekends (as discussed in Section 3.4). It is also consistent with Hassler et al. (2016) showing ratios of $[NO_x]$ / [CO] in Paris and London are higher than ratios in the LA Basin after 2010, due to larger fractions of diesel vehicles in Paris and London than in the LA Basin. Since there are recent publications of $[NO_x]$ vs. [CO], and our slopes can be compared to them, we think either way is acceptable.

We have added this discussion on [CO] vs. $[NO_x]$ in the main text at the end of the third paragraph in Section 3.4: " Hassler et al. (2016) showed the trend of $[NO_x]$ /[CO] in the Los Angeles Basin, and the ratio is between 0.1 and 0.2 after  2010, which agrees well with our results. There are also some previous studies showing ratio of [CO] / $[NO_x]$ from regions near heavy traffic emission. Parrish et al. (2012) reported the slope of [CO] vs. $[NO_x]$ was in the range of 6.3 to 18.9 for the measurement from 1987 to 1999. Wallace et al. (2012) reported the slope of [CO] vs. $[NO_x]$ was 4.2 in the morning rush hours in 2009. The slopes of [CO] vs. $[NO_x]$ in this study are 3.14 and 7.75 for weekdays and weekends, respectively.  Therefore, our results on $NO_x$ and CO are comparable with these previous studies."

Hassler, B., McDonald, B.C., Frost, G.J., Borbon, A., Carslaw, D.C., Civerolo, K., Granier, C., Monks, P.S., Monks, S., Parrish, D.D., Pollack, I.B., Rosenlof, K.H., Ryerson, T.B., von Schneidemesser, E. and Trainer, M.: Analysis of long-term observations of NOx and CO in megacities and application to constraining emissions inventories, Geophy. Res. Lett., 43, 9920-9930,doi: 10.1002/2016GL069894, 2016.

**12.     P14. The analysis of the weekend / weekday comparison of ozone is perhaps more than what the data can support. There were only 2 weekend periods. Is this really enough data to statistically demonstrate that weekends have different ozone production rates that weekdays? Isn't the production and accumulation of ozone in the airshed also affected by meteorology (irradiance,**

**dispersion)? How were these factors accounted for? You state poor statistics in explaining HCHO patterns. The weekday /weekend difference of vehicle emission on ozone production is interesting but you do not have a statistically relevant difference with 15 days of data. This should be recognized in this section. I would recommend you can place the campaign data into context with ozone data from the NAPS site for a multi-year summer period.**

Yes, we agree that only 16 days of results, including 4 weekend days, may not be statistically robust enough to show the difference on ozone production. We also think that the different levels of $NO_x$ between weekdays and weekends affect the ozone production. $O_3$ production actually goes down at high $NO_x$ (on weekdays), because the OH is reacting with $NO_2$ to form $HNO_3$ and not with VOCs to form $O_3$ and HCHO. Weekend $NO_x$ levels more efficiently form $O_3$ and HCHO.

Yes, the production and accumulation of $O_3$ in the surface air probably was also affected by meteorology. As has been discussed in the CO and $NH_3$ sections, turbulent mixing was much stronger during the day, and sensible heat flux H reached maximum during 12 to 14:00 EDT (Fig. 2). Turbulent mixing would "dilute" the pollutants accumulated in the surface air, so they would not contribute to the peak of $O_3$ in the early afternoon.

The NAPS station which provided measurement results for this manuscript was a recently installed station, so it was there only for that summer. However, there is another nearby NAPS station which has been in operation for several years. This station is situated about 175 m south of Highway 401, and about 200 m southwest of the new NAPS research trailer. We have extracted the $O_3$ data from that NAPS station for July and August from 2013 to 2015, to show some multi-year statistics.

Diurnal plots of $O_3$ on weekdays and weekends in the summer of 2013, 2014 and 2015 are shown in Fig. S7. For 2014 and 2015, the median of the diurnal average on weekends from 7 am to midnight are higher than median on weekdays, which is consistent with the results from the new research NAPS trailer shown in the main text and Fig.9. There is no resolvable difference between weekdays and weekends in 2013. Therefore, the difference between weekdays and weekends we observed in our 16-day measurement agrees with longer observations in 2014 and 2015, which suggests it is representative. We are not expecting the results from these three summers to be same as results from the research NAPS trailer, since this NAPS station is considerably farther from the highway. It is also interesting to see that for 2015, the maximum median [$O_3$] on weekends (50 ppb) from the regular NAPS station was lower than weekends median [$O_3$] (54 ppb) from the highway NAPS research trailer. Both stations show differences of 20 – 24 ppb of maximum median [$O_3$] between weekdays and weekends.

We recognize we do not have enough weekend data to be statistically robust, so as suggested, we have included this discussion above in the supplemental material, and in Fig. S7. We also have added two sentences to the main text in Section 3.4 to summarize these results and thoughts. "To evaluate how representative the contrasts between weekends and weekdays based on this 16-day data set are compared to longer timeframes, 3 summers of $O_3$ measurements from a nearby NAPS station in Toronto West were extracted and analyzed (Supplementary material Section 3 and Fig. S7). Similar diurnal patterns and differences were observed in 2 of the 3 years, suggesting that the analysis presented above is representative of longer terms as well."

**13.    P15. If the gas phase mechanism in the GEM-MACH model does not explicitly represent HCHO then it shouldn't be portrayed as HCHO in Figure 6, that is somewhat misleading if one doesn't read the fine print. What other compounds are included with HCHO, methacrolein and methyl vinyl ketone? If this is the case then it I suggest leaving out the model data in Fig 6 for "HCHO".**

Methacrolein is partially included in the HCHO lumped model species. As this reviewer suggested, HCHO GEM-MACH results have been removed from Fig. 6.

**14.    P15. The HCN section is very brief. Any idea why it is so variable; most data appear below DL of instrument except for 3 days at the end of the campaign. If HCN is from vehicle exhaust why isn't it elevated when CO was elevated? It is hard to tell from the figure, but it doesn't seem to follow CO.**

As the reviewer noted, most HCN mixing ratios were below detection limit. HCN also did not follow CO, since it was below 1 ppb for most of the period we measured. The variable HCN mixing ratio from FTIR spectrum analysis is probably due to operating so close to its detection limit, which makes it hard to fit HCN's absorption features among much more abundant interfering gases. However, indications are that we did detect a real HCN signal for at least a few short periods. We are very interested in HCN because it has adverse health effects and our colleagues have observed and modeled HCN in emissions from vehicles in the past (Moussa et al., 2016).  One sentence has been added to the text in this section: "Only on July 28, 29 and 30th, the HCN was observed above its detection limit".

**15.    P15. I don't understand the reasoning behind the statement " ...flat on weekends, indicating that a large component of CH3OH may have come from traffic emissions". Methanol doesn't co-vary with CO from examination of the figures. I find it hard to believe all CH3OH in an urban area is due to vehicles. What are other sources of methanol? As far as I know methanol is not included as a compound in vehicle emission inventories by the US EPA but perhaps this is different in Canada? Trees emit methanol. You would probably measure similar levels of methanol outside of the Toronto urban area as a result. Are there other urban sources of methanol that are relevant, solvent use for example?**

The reasoning here is comparing the diurnal cycles on weekdays and weekends; weekends did not have the early morning peak which corresponds to rush hour traffic on weekdays, so the early morning peak of $CH_3OH$ on weekdays was probably from traffic emissions. In addition, in the early morning on a few weekdays, we found linear relationship between CO and $CH_3OH$, and the slopes have been shown in Table 2 to estimate emission rate of $CH_3OH$ from traffic. The linear relationship in the early morning on weekdays also suggests at least some $CH_3OH$ was from traffic emissions.

Besides the two references in the manuscript, there are more studies on urban VOC fluxes. Rantala et al. (2016) studied urban VOC fluxes in urban Helsinki and found methanol fluxes were correlated with traffic and with CO fluxes. Traffic could partially explain the observed methanol. Sahu and Saxena (2015) also reported $CH_3OH$ mixing ratios at Ahmedabad (an urban site in India), and both traffic emission and the transport from biomass burning and biogenic sources outside the city contributed to

CH$_3$OH. Reyes et al. (2006) reported vehicle emission of non-regulated pollutants, including methanol, by using local gasoline and driving conditions in Mexico City.

Other sources of methanol include biogenic sources, such as trees and forest plants, residential wood combustion, and biomass burning. Our measurement could not directly distinguish different sources. We found no relationship of [CH$_3$OH] with ambient temperature (Fig. S6c), and we observed a linear relationship between [CH$_3$OH] and [CO] in the early morning on some weekdays. Therefore we concluded that at least some of the observed CH$_3$OH was from the traffic.

We have revised Section 3.6: "As shown in Fig. 6, mixing ratios of CH$_3$OH from the FTIR were between 2 and 20 ppb most of the time, with some high spikes. Figure 12 presents the corresponding average weekday and weekend diurnal cycles of CH$_3$OH for the study period. This plot shows the mixing ratio reached a peak (maximum of 20 ppb at 7:30) from 7:00 to 9:00 on weekdays whereas there was no peak in the mornings on weekends. In addition, a linear relationship between [CH$_3$OH] and [CO] was observed during the early morning rush hours on some weekdays (Table 2). These results suggest that at least a fraction of observed CH$_3$OH was from traffic emissions. Observations of methanol associated with traffic have been reported in other studies. Rogers et al. (2006) reported CH$_3$OH in the diluted pipeline exhaust of a mobile laboratory. CH$_3$OH may also come from non-engine sources, such as windshield wiper fluid. Durant et al. (2010) measured gas and particle pollutants near Interstate 93 in Massachusetts. They reported CH$_3$OH was above 20 ppb at 7:20 50 m downwind of the highway, possibly with contributions from some other local sources. Reyes et al. (2006) reported vehicle emission of non-regulated pollutants, including methanol, by using local gasoline and driving conditions in Mexico City. Rantala et al. (2016) studied urban VOC fluxes in urban Helsinki and found methanol fluxes were correlated with traffic and with CO fluxes, traffic could partially explain the observed methanol. Sahu and Saxena (2015) also reported CH$_3$OH mixing ratios at Ahmedabad (an urban site in India), and both traffic emission and the transport from biomass burning and biogenic sources outside the city contributed to CH$_3$OH. The mixing ratio of CH$_3$OH we observed did not correlate with ambient temperature (Fig. S6c), so there was no strong indication of biogenic sources."

Rantala, P., Järvi, L., Taipale, R., Laurila, T.K., Patokoski, J., Kajos, M.K., Kurppa, M., Haapanala, S., Siivola, E., Petäjä, T., Ruuskanen, T.M. and Rinne, J.: Anthropogenic and biogenic influence on VOC fluxes at an urban background site in Helsinki, Finland, Atmos. Chem. Phys., 16, 7981-8007,doi: 10.5194/acp-16-7981-2016, 2016.

Sahu, L.K. and Saxena, P.: High time and mass resolved PTR-TOF-MS measurements of VOCs at an urban site of India during winter: Role of anthropogenic, biomass burning, biogenic and photochemical sources, Atmos. Res., 164-165, 84-94,doi: 10.1016/j.atmosres.2015.04.021, 2015.

Reyes, F., Grutter, M., Jazcilevich, A. and González-Oropeza, R.: Tecnical Note: Analysis of non-regulated vehicular emissions by extractive FTIR spectrometry: Tests on a hybrid car in Mexico City, Atmos. Chem. Phys., 6, 5339-5346, 2006.